# Chemically-induced degradation of the endoplasmic-reticulum stress sensor IRE1 by a VHL-recruiting chimera

Jin Du [1], Elisia Villemure[2,6], Matthew Johnson[3,6], Caleigh Azumaya[3], Catarina J. Gaspar[1], Scot Marsters[1], David Lawrence[1], Scott Foster[1], Alexis Rohou [3], Tommy K. Cheung[4], Christopher M. Rose[4], Thomas Garner[5], Soo Ro[5], Kevin Clark[5], Maureen H. Beresini[5], Marie-Gabrielle Braun[2], Joachim Rudolph [2] ✉, Peter Hsu [3] ✉ & Avi Ashkenazi [1] ✉

The endoplasmic-reticulum (ER) transmembrane protein IRE1 mitigates ER stress through kinase-endoribonuclease and scaffolding activities. Cancer cells often co-opt IRE1 to facilitate growth. An IRE1-RNase inhibitor has entered clinical trials; however, recent work uncovered a significant nonenzymatic IRE1 dependency in cancer. To fully disrupt IRE1, we describe a proteolysis-targeting chimera (G6374) that couples an IRE1-kinase ligand to a compound that binds the ubiquitin Cullin-RING Ligase (CRL) substrate receptor, VHL. G6374 induces a stable, cooperative interaction between IRE1 and VHL, driving K48-linked ubiquitination on two principal lysine residues in the IRE1-kinase domain and inducing proteasomal IRE1 degradation. Cryogenic electron microscopy and mutagenesis studies reveal a 2:2 IRE1:VHL ternary-complex topology and critical interactional features, informing future designs. G6374 blocks growth of IRE1-dependent cancer cells irrespective of their dependency mode, while sparing IRE1-independent cells. We provide a proof-of-concept for VHL-based degradation of an ER-transmembrane protein, advancing strategies to fully disrupt IRE1.

Inositol-requiring enzyme 1α (IRE1α, herein IRE1) plays a central role in mediating the unfolded protein response (UPR), which enables eukaryotic cells to maintain protein-folding homeostasis of their endoplasmic reticulum (ER)[1–3]. IRE1 comprises an N-terminal ER-lumenal domain and a single-pass transmembrane domain, both of which sense ER stress through direct and indirect mechanisms[1–3]. In its C-terminal cytoplasmic region, IRE1 possesses a tandem kinase-RNase (KR) module, which transduces several intracellular functions downstream of IRE1 activation. Upon detecting ER stress, steady-state IRE1 homodimers undergo further oligomerization, in conjunction with

*trans* autophosphorylation of the kinase domain and allosteric stimulation of RNase activity[4–7]. The IRE1 RNase performs two cardinal functions: (1) activation of the transcription factor XBP1s, through mRNA-intron excision followed by RtcB-mediated nonconventional splicing[3,8]; (2) cleavage of multiple cellular mRNAs and miRNAs via an endomotif-directed process called RIDD (for regulated IRE1-dependent decay)[9], and a less stringent mechanism dubbed RIDDLE (for RIDD-lacking endomotif)[6]. XBP1s transcriptionally reprograms the cell to increase ER-based protein-folding capacity and to promote ER-associated degradation (ERAD) of misfolded proteins. RIDD and RID-

[1]Department of Research Oncology, Genentech, Inc., South San Francisco, CA, USA. [2]Department of Discovery Chemistry, Genentech, Inc., South San Francisco, CA, USA. [3]Department of Structural Biology, Genentech, Inc., South San Francisco, CA, USA. [4]Department of Proteomic and Genomic Technologies, Genentech, Inc., South San Francisco, CA, USA. [5]Department of Biochemical and Cellular Pharmacology, Genentech, Inc., South San Francisco, CA, USA. [6]These authors contributed equally: Elisia Villemure, Matthew Johnson. ✉e-mail: joachimr@gene.com; hsup2@gene.com; aa@gene.com

DLE help to abate the ER's protein-folding load, and can regulate additional cellular functions, including lipid synthesis[10], lysosome repositioning[11], apoptosis[12,13], and cellular fitness[6].

Cancer cells can co-opt IRE1's cytoprotective functions to circumvent ER stress, driven by intrinsic and extrinsic conditions in the tumor microenvironment, while sustaining malignant growth and evading immunity[3,14,15]. Cancer dependency on IRE1's enzymatic function is well established, and small-molecule kinase and RNase inhibitors of IRE1 have been investigated in multiple preclinical models[3,16]. The IRE1 RNase inhibitor ORIN1001 has recently advanced into clinical trials[17]. Importantly, however, recent work has revealed that certain cancer cells display a nonenzymatic dependency on IRE1, indicating that scaffolding function(s) of IRE1, several of which have been reported, can be crucial for tumor growth[18,19]. In such instances, disruption of IRE1's enzymatic activity by kinase and/or RNase inhibitors failed to block tumor growth, whereas on-target IRE1 depletion via RNA interference led to strong tumor regression[18,19]. We therefore reasoned that a strategy to chemically induce substantive degradation of the IRE1 protein might enable its more comprehensive disruption to inhibit growth of IRE1-dependent cancer cells regardless of their mode of IRE1 dependency.

Toward this latter goal, we designed and synthesized a proteolysis-targeting chimera (PROTAC) aimed to deplete IRE1 by inducing its polyubiquitination and proteasomal degradation. Heterobifunctional degraders typically consist of two ligand moieties joined by a suitable linker: one ligand binds to the targeted protein of interest, while the other binds to an E3 ubiquitin ligase expected to ubiquitinate the substrate and thereby drive its degradation[20,21]. An IRE1-targeting PROTAC has been previously reported[22,23]. However, this compound, which is based on an IRE1 ligand that binds to the RNase domain linked to a ligand that recruits the ubiquitin ligase CRBN, afforded only partial degradation of IRE1 in HEK293T cells, while its ability to inhibit IRE1-dependent cancer-cell growth was not investigated[22,23]. To obtain a degrader that could deplete IRE1 more fully, we took advantage of the diversity of previously identified ligands that bind specifically and tightly to the ATP pocket of IRE1's kinase domain[24-26]. In concert, we leveraged a previously established ligand that recognizes the von Hippel-Lindau (VHL) E3 ligase with high affinity[27,28]. VHL is a substrate receptor component of a multi-subunit Cullin-RING ligase (CRL) complex comprised of CUL2, Elongin B, Elongin C, and RBX1, which recruits a ubiquitin-charged E2 enzyme to build polyubiquitin chains[29,30]. Like other CRLs, the CRL2[VHL] complex is activated through post-translational modification of CUL2 by neddylation[31]. Here, we describe a heterobifunctional degrader that promotes efficient K48-linked ubiquitination and complete depletion of endogenous IRE1 through VHL in cancer cells. By solving the structure of the IRE1-degrader-VHL ternary complex via cryogenic electron microscopy (cryoEM), together with performing mutational functional studies, we obtain key mechanistic insight into the overall organization and specific interactions of the assembly. Furthermore, we demonstrate that VHL-based degradation of IRE1 selectively blocks growth of IRE1-dependent cancer cell lines regardless of their mode of dependency on IRE1, while sparing IRE1-independent cells. This work demonstrates effective chemically-induced VHL-based degradation for an ER-transmembrane protein. Moreover, our findings establish a promising strategy to fully disrupt IRE1 for therapeutic purposes in cancer and beyond.

## Results

### Designing an effective degrader for endogenous IRE1 in cancer cells

To induce IRE1-protein degradation, we designed and synthesized a panel of compounds that link diverse IRE1-kinase ligands[16] to an established VHL ligand[27,28]. As a conceptual starting point for designing bivalent degraders, we used Ex 91 (Supplementary Fig. 1a)—a potent

and selective dual IRE1α KR inhibitor[32]. Since RNase inhibition is not essential for degradation, we envisioned that truncating Ex 91's RNase allosteric control element[25] (highlighted in blue in Supplementary Fig. 1a) would increase the probability of identifying molecules with better drug-like properties. This led to G1167—still a potent IRE1 inhibitor with reduced polarity and half the molecular weight of Ex 91—as a further advancement. We then generated a library of ~100 bivalent compounds using a piperazine exit vector with linkers of varied length and composition (alkyl and PEG chains), in combination with ligands that recognize different E3 ligases, namely, VHL, CRBN, and XIAP. Based on a high-throughput degradation assay (see Methods), we elected to focus on the VHL-based IRE1 PROTACs. To facilitate IRE1 accessibility to CRL2[VHL] complexes and enhance ubiquitin transfer, we identified a four-carbon alkyl linker that provides optimal spacing and flexibility. Among the synthesized compounds, G6374 (Fig. 1a) stood out as particularly effective at depleting endogenous cellular IRE1. We compared the effectiveness of G6374 with the previously published IRE1 degrader CPD-2828, which links an IRE1-RNase ligand to a CRBN ligand[22,23]. Immunoblot (IB) analysis of AMO1 multiple myeloma (MM) cells, which express relatively high levels of IRE1[18], demonstrated that G6374 induced more potent and substantial depletion of the IRE1 protein as compared to CPD-2828, with a respective $D_{max}$ of 93 versus 71%, and a $DC_{50}$ of 0.04 versus 0.64 μM (Fig. 1b). Analysis of IRE1 degradation in AMO1 cells over a time course indicated that G6374 induces near complete IRE1 depletion by 4 h whereas CPD-2828 achieved only partial depletion by 8 h (Supplementary Fig. 1b), confirming more efficient IRE1 degradation by G6374. The G6374 compound also induced effective depletion of IRE1 in KMS27 MM cells, which express similarly high IRE1 levels, with a $D_{max}$ of 98% and a $DC_{50}$ of 0.13 μM (Supplementary Fig. 1c). Pharmacological induction of ER stress by thapsigargin or tunicamycin did not encumber G6374-driven IRE1 depletion in either cell line (Supplementary Fig. 1d), demonstrating the compound's capacity to induce IRE1 degradation even in the context of significant ER disruption.

To assess the selectivity of G6374 for IRE1 as compared to other kinases, we tested the compound in vitro at 1 μM against a panel of 220 kinases. This analysis identified 9 other kinases that were inhibited by >90%, besides IRE1, which showed complete inhibition (Fig. 1c). Importantly, IB analysis of AMO1 and KMS27 cells treated with G6374 at 0.1, 0.3 and 1 μM using validated available antibodies to all nine of these kinases showed that while JNK3 was not detectable, the other eight kinases were not appreciably depleted by the compound (Fig. 1d). These results are consistent with the expectation that compound-mediated recruitment of VHL to the substrate in a manner that enables efficient ubiquitin transfer and degradation has more complex requirements than does compound binding to the substrate. Moreover, global proteomics analysis of AMO1 cells treated with G6374, as compared to G2642—an epimer control compound that contains a *cis*-VHL ligand and does not degrade cellular IRE1 (Supplementary Fig. 1e, f)—reaffirmed IRE1 as the most substantially and significantly depleted protein target of G6374 (Fig. 1e).

CRL-mediated protein degradation requires the function of an E1 ubiquitin-activating enzyme (E1), as well as modification of the scaffolding Cullin protein by neddylation[31,33]. Both an E1 inhibitor and a neddylation inhibitor blocked G6374-induced IRE1 protein depletion in AMO1 cells (Fig. 1f), consistent with the involvement of CRL2[VHL] complexes. Moreover, whereas lysosome inhibition with bafilomycin A1 or chloroquine did not inhibit G6374-induced IRE1 depletion, proteasome inhibition with bortezomib significantly preserved IRE1 (Fig. 1g). Furthermore, inhibition of p97/VCP—known to be required for ERAD[34]—only slightly impaired G6374-induced IRE1 depletion (Supplementary Fig. 1g), suggesting minimal involvement of ERAD. Of note, siRNA-based knockdown of VHL, performed in HCT116 cells because they are more efficiently transfected than AMO1 cells, abolished G6374-induced IRE1 depletion (Supplementary Fig. 1h),

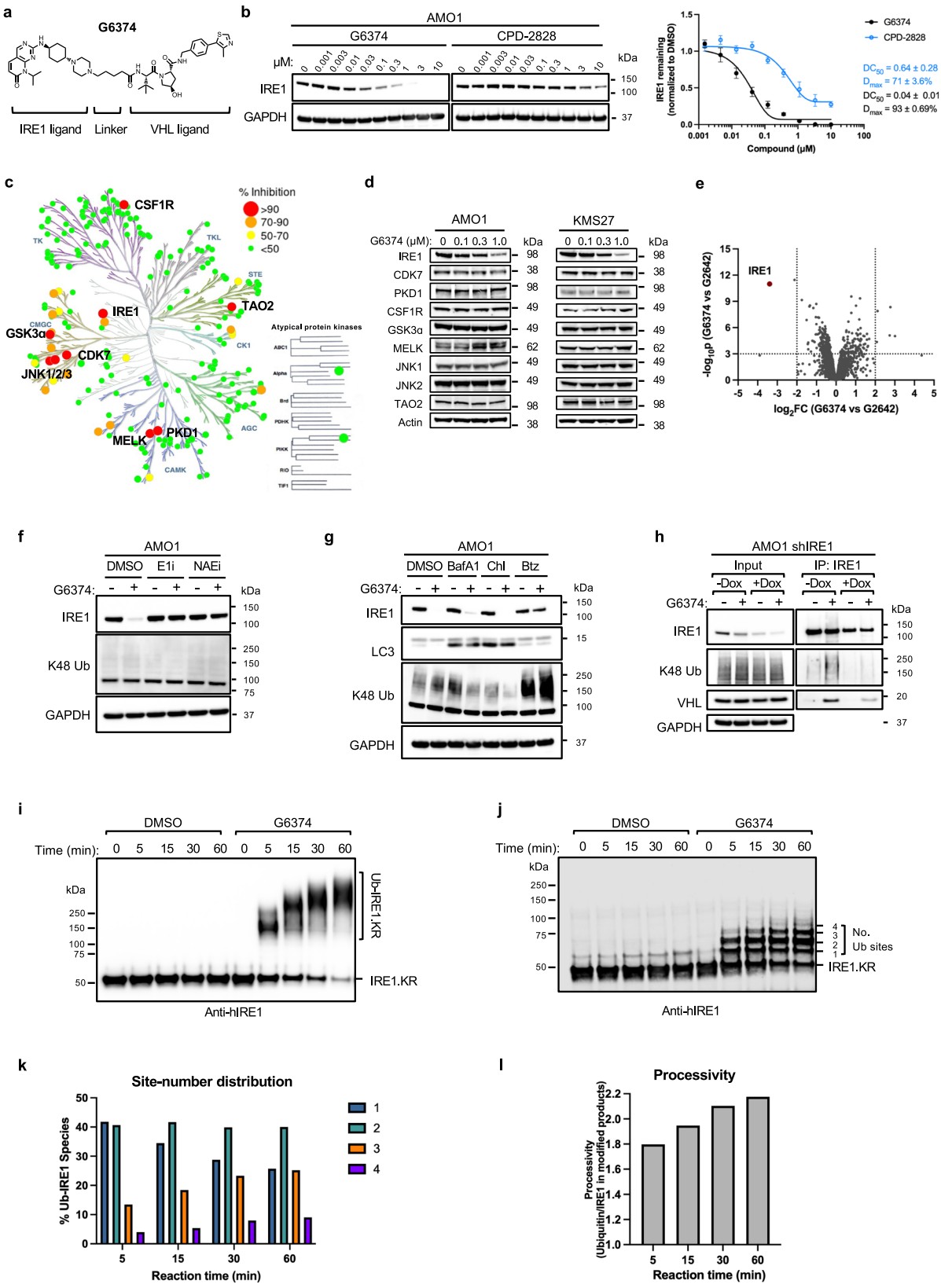

confirming the dependence on VHL. Together, these results suggest that G6374-driven IRE1 degradation requires CRL2$^{VHL}$-based ubiquitin transfer and proteasome activity.

To investigate the induction of IRE1 ubiquitination more directly, we performed an immunoprecipitation (IP) of the IRE1 protein using a validated IRE1-specific antibody. We then visualized covalently attached ubiquitin (Ub) on IRE1 by IB with a validated K48-Ub-specific antibody—thus detecting a modification that typically directs substrate proteins to the proteasome for degradation[35,36]. As compared to DMSO, G6374 induced a significant increase in K48-linked ubiquitination of endogenous IRE1 in AMO1 cells (Fig. 1h). Doxycycline (Dox)-induced shRNA-mediated silencing of IRE1[37] decreased both the

**Fig. 1 | G6374 induces efficient ubiquitination and degradation of IRE1.**
**a** Chemical structure of the heterobifunctional IRE1 degrader G6374. **b** Immunoblot (IB) analysis of endogenous IRE1 depletion in AMO1 cells (17 h) with a titration of G6374 compared to CPD-2828 (left) and IB quantification (right; $n = 4$ biological replicates, mean ± SEM). **c** Selectivity of G6374 (1 μM, 1 h) determined by SelectScreen Profiling against 220 kinases. **d** IB assessment of kinases inhibited by 90% or more for depletion by G6374 (4 h) in AMO1 and KMS27 cells. **e** Global proteomics analysis of G6374- versus G2642-induced protein depletion (1 μM, 4 h) in AMO1 cells. **f** IRE1 depletion in AMO1 cells treated with G6374 (3 μM, 2 h), with or without E1 inhibitor TAK-243 (E1i; 2 μM) or Neddylation inhibitor TAS4464 (NAEi; 0.2 μM). E1i and NAEi were added 4 h prior to G6374. Samples were analyzed by IB for IRE1 or K48-linked ubiquitin (Ub). **g** IRE1 depletion by G6374 (3 μM, 2 h) in AMO1 cells treated with bafilomycin A1 (BafA1; 30 nM), chloroquine (Chl; 100 μM), or bortezomib (Btz; 100 nM), added 2 h prior to G6374. Samples were analyzed by

IB for IRE1, and respectively for LC3 or K48-Ub to confirm lysosome or proteasome inhibition. **h** Immunoprecipitation (IP) of endogenous IRE1 from AMO1 cells expressing doxycycline (Dox)-inducible IRE1 shRNA, treated with G6374 (3 μM, 2 h) or DMSO. Dox-induced IRE1 depletion controls for IP-IB specificity. **i** In vitro ubiquitination of recombinant IRE1.KR incubated with G6374 plus recombinant components of the CRL2$^{VHL}$ complex and native ubiquitin. Samples were analyzed by IRE1 IB. **j** In vitro ubiquitination of recombinant IRE1.KR as in (**i**) but using Me-Ub instead of Ub. Samples were analyzed by IRE1 IB. The number of Ub sites is based on relative molecular mass. **k** Distribution of IRE1 protein with 1, 2, 3, or 4 attached Me-Ub, quantified from (**j**). "% Ub-IRE1 species" was calculated from the band intensity of each species divided by the sum of IRE1 Ub1, Ub2, Ub3, and Ub4 intensities. **l** Reaction processivity calculated from (**j**) as the probability-weighted average Ub number in each respective species.

baseline K48-Ub signal and its elevation by IRE1 modification (Fig. 1h), confirming specific detection of K48-Ub on IRE1 by this IP-IB method.

To further examine whether G6374 could directly promote IRE1 ubiquitination by CRL2$^{VHL}$, we incubated a recombinant protein comprising the human IRE1 KR domain (IRE1.KR)[6,25] with a recombinant CRL2$^{VHL}$ complex consisting of VHL, CUL2, EloB, EloC, and RBX1, plus Ub[38]. In this setting, we observed clear G6374-dependent IRE1 poly-ubiquitination (Fig. 1i), indicating direct and processive G6374-driven Ub transfer by CRL2$^{VHL}$ to IRE1. Of note, both nonphosphorylated (0P) and phosphorylated (3P)[25] IRE1.KR (confirmed by LC-MS, Supplementary Fig. 1i) underwent comparable G6374-induced ubiquitination (Supplementary Fig. 1j), suggesting that Ub transfer by CRL2$^{VHL}$ to IRE1 can proceed independent of phosphorylation status on IRE1's kinase-activation loop.

To determine the number of ubiquitination sites that could be modified by CRL2$^{VHL}$ on IRE1, we performed the reaction with methylated ubiquitin (Me-Ub), which permits mono-Ub transfer to each given site but prohibits Ub-chain elongation[39]. In the presence of G6374, CRL2$^{VHL}$ rapidly ubiquitinated two sites on IRE1.KR, followed by slower Me-Ub transfer to a third site, and a less efficient yet detectable Me-Ub addition to a fourth site (Fig. 1j). Quantification of the IB and calculation of the reaction's processivity[40,41] indicated that CRL2$^{VHL}$ mainly modified two of the four detectable sites on IRE1 (Fig. 1k, l). To further analyze the modification of IRE1.KR, we used mass spectrometry (MS), which confirmed that the predominant Ub linkage was K48 (Supplementary Fig. 1k, l). Moreover, MS analysis identified K704 and K717—located in the kinase domain—as predominant ubiquitination sites (Supplementary Fig. 1m-o). We could not ascertain the location of the other two potential ubiquitination sites by MS. Toward further validation, we generated and purified three IRE1 mutants: K704R, K717R, and K704R/K717R. While these mutations did not abolish IRE1 ubiquitination, their ubiquitination-site patterns were different from that of WT IRE1, suggesting unmasking of secondary positions for ubiquitin conjugation in the mutants (Supplementary Fig. 1p, q). Together, these results show that G6374 drives CRL2$^{VHL}$-mediated K48-linked polyubiquitination of IRE1, directed primarily to two lysine sites in the kinase domain.

Our initial experiments employed UBE2R1 as the E2 ubiquitin-conjugating enzyme, based on its known utility in the validation of VHL-based PROTAC function[38]. Whereas UBE2R1 has a stronger inherent tendency to generate K48 linkages, the E2 enzymes UBE2D1 and UBE2D2 have a broader linkage propensity that is heavily influenced by the E3 ligase partner[42-44]. To evaluate the effect of different E2 ubiquitin-conjugating enzymes on IRE1 ubiquitination, we also tested the activity of UBE2D1 and UBE2D2 in the in vitro assays. Unlike UBE2R1, UBE2D1 and UBE2D2 predominantly generated K11-linked polyubiquitination of IRE1.KR (Supplementary Fig. 1r). Despite the divergent linkage specificity of these different E2 enzymes, Me-Ub experiments coupled with LC-MS/MS analysis showed that the key IRE1 lysine (K704) targeted by G6374-mediated ubiquitination remained

unchanged (Supplementary Fig. 1s–u). Together, these results suggest that K704 is the most accessible ubiquitin-acceptor in this PROTAC-mediated complex.

## G6374 promotes a stable association between IRE1 and VHL

To examine whether G6374 could induce a stable interaction between cellular IRE1 and VHL, we performed an IP of endogenous IRE1 from AMO1 cells and revealed associated endogenous VHL protein by IB. We could detect a weak IRE1:VHL interaction at baseline; however, G6374 induced a much stronger interaction between the two proteins, in concert with IRE1 depletion (Fig. 2a). Co-IP analysis of recombinant IRE1.KR and VHL:EloC:EloB (VCB) proteins further confirmed their ability to associate minimally in the absence of G6374 and markedly more strongly in its presence (Supplementary Fig. 2a). Importantly, addition of the highly selective, ATP-competitive IRE1-kinase inhibitor G5758[26] in excess of G6374 attenuated the induced association between IRE1 and VHL as well as the depletion of IRE1 (Fig. 2a), demonstrating that the recruitment of VHL to IRE1 is both specific and functionally productive.

To monitor the G6374-induced IRE1:VHL interaction more quantitatively, we established a Nanoluciferase Bioluminescence Resonance Energy Transfer (NanoBRET) assay[45]. We expressed cDNA constructs encoding IRE1 with a C-terminal NanoLuc Luciferase tag, and VHL with an N-terminal Halo-tag. To avoid interference by the endogenous IRE1 protein, we chose HEK293T cells, which express low levels of IRE1 (Supplementary Fig. 2b). IB analysis confirmed that G6374 addition depleted the over-expressed IRE1-NanoLuc protein in a concentration-dependent manner (Supplementary Fig. 2c). The NanoBRET signal ratio, normalized to the amount of IRE1 at each G6374 concentration, indicated a concentration-dependent interaction between IRE1 and VHL, with an estimated EC$_{50}$ of 110 nM (Fig. 2b). In concert, the NanoLuc luminescence signal showed that G6374 induced a concentration-dependent depletion of the IRE1 protein, with an estimated DC$_{50}$ of 130 nM (Fig. 2c), comparable to the DC$_{50}$ for endogenous IRE1 in AMO1 and KMS27 cells. In keeping with the IP-IB results in AMO1 cells (Fig. 2a), the NanoBRET analysis in HEK293T cells confirmed that the IRE1 kinase inhibitor G5758 could disrupt G6374-induced interaction between IRE1 and VHL (Supplementary Fig. 2d). These results validate our NanoBRET assay and further support the conclusion that G6374 induces a stable association between IRE1 and VHL and drives IRE1 degradation.

We next tested whether the IRE1:VHL interaction can be recapitulated by purified recombinant IRE1 and VHL proteins. We incubated purified IRE1.KR with VHL, pre-complexed with EloB and EloC—known to stabilize VHL[46,47]. Size-exclusion chromatography revealed that G6374 induced a single, monodispersed peak of an IRE1:VHL:EloC:EloB complex (Fig. 2d, e), demonstrating the assembly of a stable ternary complex in vitro.

To further characterize the binary and ternary interactions between G6374, IRE1, and VHL, we performed surface plasmon

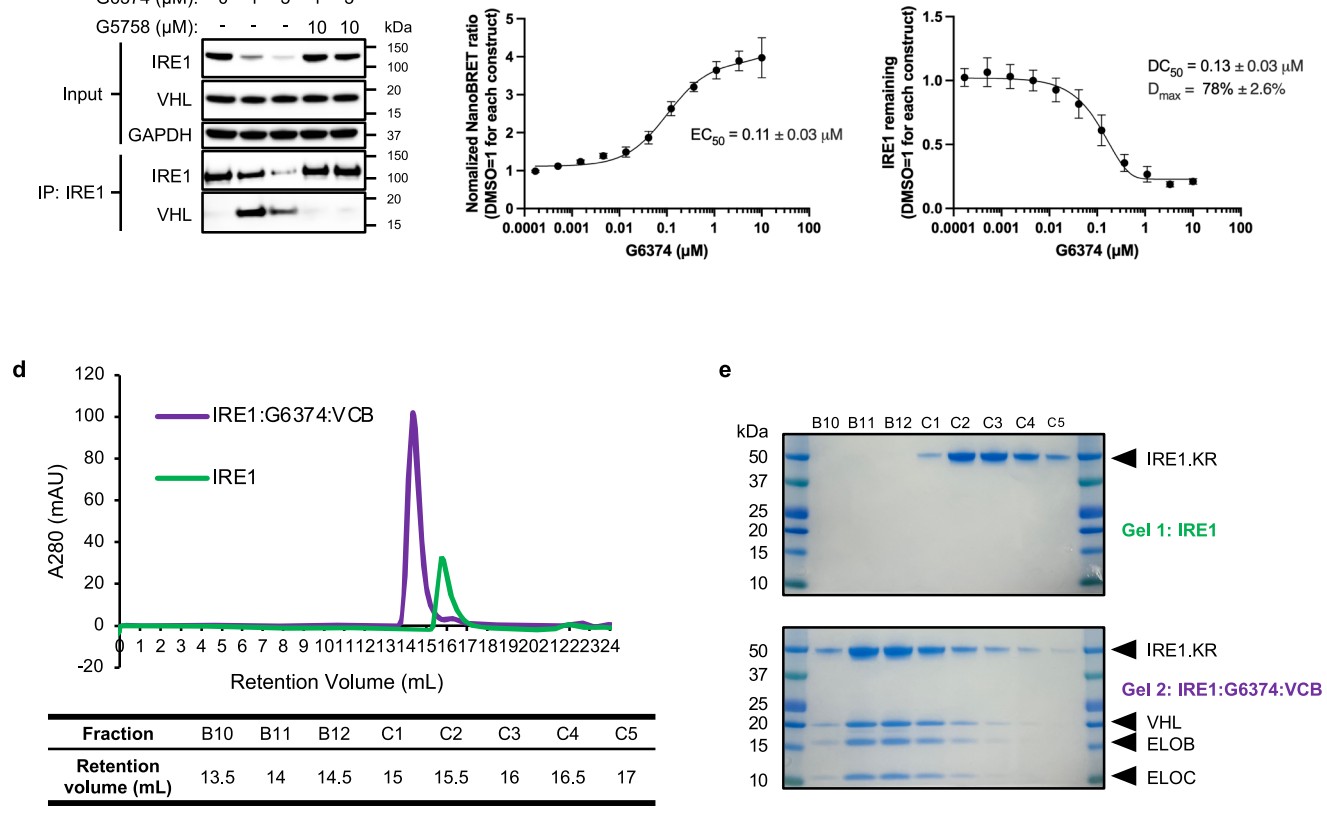

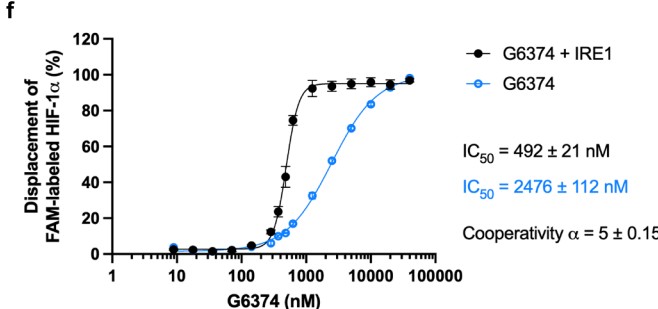

**Fig. 2 | G6374 promotes a stable association between IRE1 and VHL. a** co-IP of endogenous VHL with endogenous IRE1 from AMO1 cells upon addition of a titration of G6374 and competition with an IRE1 kinase inhibitor, G5758. Samples were analyzed by IB for IRE1, VHL, and GAPDH. **b, c** NanoBRET assay with over-expressed IRE1 and VHL in HEK293T cells with a titration of G6374 ($n = 3$ biological replicates, mean ± SEM). $EC_{50}$ indicates the degrader concentration that achieves 50% maximal binding (as a readout of the NanoBRET ratio) (**b**). $DC_{50}$ indicates the concentration at which 50% of the maximal achievable IRE1 depletion is observed; $D_{max}$ indicates the maximal percent depletion of IRE1 as compared to baseline (as a readout of the NanoLuc signal) (**c**). **d** Size-exclusion chromatography (SEC) of the IRE1:G6374:VCB complex or IRE1 alone. **e** Coomassie Blue staining of fractions from the peaks in (**d**) on SDS-PAGE gels. **f** Fluorescence polarization displacement assay performed with a titration of G6374 in the absence or presence of saturating IRE1 concentration (2.4 μM) to VCB bound with FAM-labeled HIF-1α peptide ($n = 3$ biological replicates, mean ± SEM).

resonance (SPR) and fluorescence polarization (FP) analyses. SPR detected only negligible interaction between IRE1 and VHL in the absence of G6374 ($K_d > 10$ μM), while binary interaction was more efficient for IRE1:G6374 ($K_d = 0.65$ nM) than VHL:G6374 ($K_d = 83.70$ nM) (Supplementary Fig. 2e–i; Supplementary Table 1). For ternary complex cooperativity measurement, we considered that IRE1 can form dimers[4,5,7,48], which may bias immobilization-based SPR analysis. We therefore turned to a solution-based FP approach, which has previously been used to characterize cooperativity for a KRAS PROTAC[49]. To this end, we measured G6374's ability to displace a VHL-binding HIF-1α probe in the absence or presence of IRE1. Addition of

IRE1 markedly augmented this displacement, enabling half-maximal competition at a 5-fold lower concentration of G6374 (Fig. 2f; Supplementary Table 1). These results suggest that while G6374 binds more tightly to IRE1 than VHL, it induces ternary complex formation with IRE1 and VHL in a cooperative manner.

**Structural basis of IRE1 degradation by VHL:G6374**

To understand the structural basis for VHL:G6374-mediated IRE1 degradation, we studied the heteropentameric VHL:EloB:EloC:G6374:IRE1 complex by cryoEM to an average resolution of 2.6 Å (Fig. 3a; and Supplementary Fig. 3a–k, Supplementary Table 2). Our

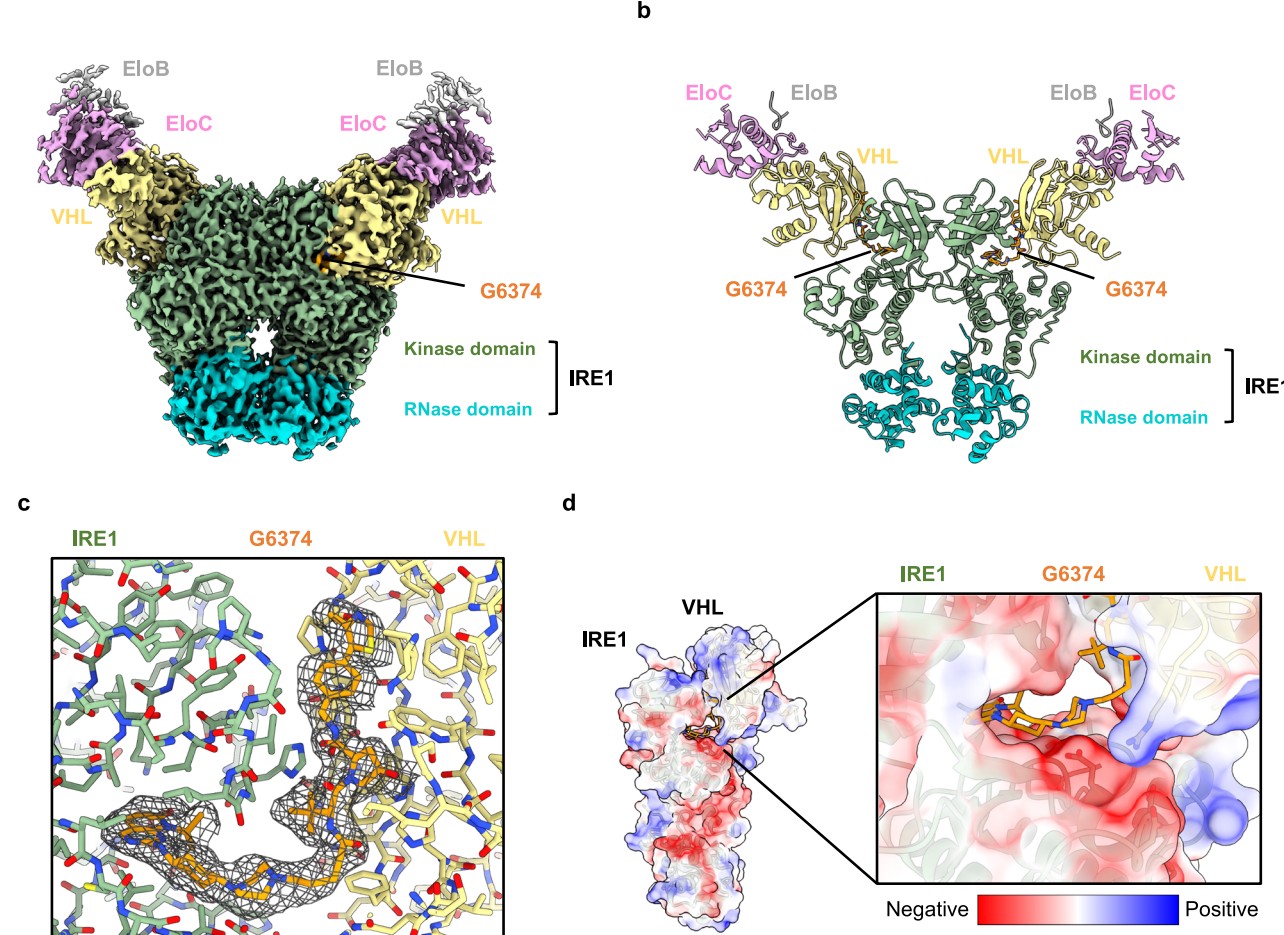

**Fig. 3 | CryoEM structural analysis reveals the 3D organization of the IRE1:G6374:VHL ternary complex. a** CryoEM map of the IRE1:G6374:VCB complex with each component color-coded. **b** Model of the IRE1:G6374:VHL complex built based on the cryoEM map. **c** Zoomed-in view of the interface showing the atomic model (stick representation) and an isosurface rendering of the cryoEM map (mesh representation). **d** Surface electrostatics of IRE1 and VHL with a zoomed-in view of the IRE1:G6374:VHL interface.

initial attempts to prepare cryoEM grids were performed without crosslinking by BS3 [bis(sulfosuccinimidyl)suberate] and addition of cetyltrimethylammonium bromide (CTAB), which led to unstable complexes and/or biased particle orientations. However, we found that BS3 crosslinking improved complex integrity on the grid, while CTAB addition circumvented preferred orientation issues. With these improvements, cryoEM analysis revealed that VHL engages the classical back-to-back (B2B) homodimeric state of IRE1[4,48], forming a 2:2 stoichiometry. The overall assembly adopts a butterfly-shaped structure, where the IRE1 kinase domain resembles the "body" and "forewings," and the RNase domain reflects the lower "hindwings." VHL:EloC forms the tips of the forewings at the edge of the complex (Fig. 3a, b). Because the EloB density had insufficient quality for de novo modeling, we built only residues 68 to 73 of this protein into the structure. G6374 resides at the interface between VHL and IRE1 (Fig. 3c), and the high local resolution (Supplementary Fig. 3d) enables its unambiguous modeling into the structure.

VHL associates via G6374 with the N-lobe of the IRE1-kinase domain in the forewings, with a total buried surface area (BSA) of ~300 Å²−an interface that is comparable to known VHL:bromodomain:PROTAC assemblies[50,51] (Supplementary Table 3). Surface electrostatics of VHL and IRE1 show the docking of an electronegative area on IRE1 into a positive patch on VHL, indicative of a complementary and productive protein-protein interaction induced by the compound (Fig. 3d).

Comparison with existing crystal structures of B2B IRE1 shows that engagement by VHL does not perturb the B2B dimer assembly of IRE1 (Supplementary Fig. 3l). In contrast, alignment of our structure with a face-to-face (F2F) dimeric IRE1 structure[52] suggests that recruitment of VHL by G6374 would sterically block the ability of IRE1 to undergo *trans*-autophosphorylation (Supplementary Fig. 3m). Thus, VHL:G6374 likely targets the basal steady-state mode[7] of the kinase for degradation.

### Details of the VHL:G6374:IRE1 interface

Like previously determined structures of VHL:PROTAC:neosubstrate complexes[50,51] (Supplementary Fig. 4a), the two ends of G6374 engage their respective targets while the four-carbon alkyl linker folds in ~90°, helping to form an overall L-shape of the molecule. The IRE1-targeting component of G6374 binds into the ATP pocket of the kinase domain[4,48], much like classical ATP-competitive inhibitors. Binding of G6374 to IRE1 does not disrupt the known salt-bridge between Lys599 and Glu612 (Supplementary Fig. 4b), which keeps the αC-helix of the IRE1-kinase domain pointing inward, thus permitting an active configuration (Supplementary Fig. 4c)[53]. Hydrophobic interactions between the pyridopyrimidinone of G6374 and the side chains of IRE1 residues Leu577, Val586, Ala597, Leu644, and Leu695 help to anchor the compound within the active site of IRE1. Additional hydrogen bond interactions between G6374 and the backbone-amide of the hinge residue Cys645 and the side chain of Glu651 also contribute

prominently to the G6374:IRE1 interaction (Fig. 4a). To assess the individual contribution of these specific interactions, we generated point mutants of IRE1 having amino acid substitutions that would disrupt either hydrophobic interactions, i.e., L577D, V586D, A597D, L644D, L695D; or hydrogen-bond formation, i.e., E651A. IB analysis confirmed that all mutants were expressed at similar levels as WT in cells (Supplementary Fig. 4d). NanoBRET analysis showed that each one of the mutations was sufficient to abolish the G6374-induced interaction between IRE1 and VHL (Fig. 4b) as well as IRE1 degradation (Fig. 4c). These results demonstrate that efficient VHL recruitment and subsequent IRE1 depletion require an extensive set of interactions between G6374 and IRE1.

At the IRE1:VHL interface, the imidazole of the His579 side chain on IRE1 forms a π-stack with the aromatic ring of VHL-Tyr98, which could potentially contribute to a G6374-independent protein-protein interface between IRE1 and VHL (Fig. 4d). Indeed, substitution of IRE1-His579 by alanine (H579A) led to a weaker IRE1:VHL interaction (Fig. 4e) and markedly diminished IRE1 degradation (Fig. 4f). IP-IB analysis revealed that disruption of the interaction of IRE1-His579 with VHL-Tyr98 effectively abolished G6374-mediated K48 ubiquitination of IRE1 (Fig. 4g, h), which may explain the substantially reduced degradation of the IRE1-H579A mutant. Sequence alignment of IRE1 against the closest potential off-target kinases of G6374 shows that His579 is unique to IRE1 and TAO2 (Fig. 1c; Supplementary Fig. 4e), suggesting that this residue contributes to the relative selectivity of G6374 for IRE1. Nevertheless, G6374 did not deplete TAO2 (Fig. 1d), suggesting that further determinants of selectivity exist besides His579. Taken together, our results demonstrate that the effective depletion of IRE1 requires protein-protein interactions as well as productive ubiquitination.

### G6374 can target monomeric IRE1 for degradation

The 2:2 configuration of the IRE1-VHL complex prompted us to explore whether CRL2[VHL] could also ubiquitinate nondimeric IRE1. Leveraging published structures[54], we modeled the full CRL2[VHL] assembly in association with IRE1.KR in its monomeric state (Fig. 5a). This model predicted that the RBX1 component of CRL2[VHL] could potentially transfer ubiquitin at least to one of the favored sites on the immediately bound IRE1 protomer, e.g., K704. To examine this further, we expressed an IRE1 variant incorporating two mutational substitutions designed to disrupt the B2B dimer, i.e., E621A and E836A (Fig. 5b). Crosslinking of IRE1 indicated that incorporating these two mutations indeed decreased dimerization (Supplementary Fig. 5a). IP-IB analysis further showed that G6374-induced ubiquitination of the double mutant was not significantly diminished as compared to that of wild-type (WT) IRE1 (Fig. 5b). In keeping with this observation, modeling the full CRL2[VHL] assembly in complex with a B2B dimer suggests the possibility of crossover ubiquitination by CRL2[VHL] on the distal, indirectly bound IRE1 protomer in addition to ubiquitination on the directly bound protomer (Supplementary Fig. 5b). Indeed, our dimeric modeling suggests that IRE1-K717 would be inaccessible to ubiquitination in the same protomer where K704 is targeted (bottom CRL2[VHL] in Supplementary Fig. 5c), since it is on the opposite side of the kinase domain, distal to RBX1-E2. However, due to the anti-parallel B2B architecture, K717 is accessible for ubiquitination to the symmetry-related CRL2[VHL] (top CRL2[VHL] in Supplementary Fig. 5c). Nevertheless, NanoBRET analysis indicated that the B2B-disrupted mutant could bind VHL and undergo degradation similarly to WT IRE1 (Fig. 5c, d). These results suggest that G6374 drives comparable ubiquitination and degradation of both IRE1 configurations.

### G6374 selectively inhibits IRE1-dependent cancer-cell proliferation regardless of the underlying dependency mode

To examine whether G6374 could be leveraged against cancer cells that possess either enzymatic or nonenzymatic IRE1 dependency, we tested the effect of the compound as compared to Dox-induced shIRE1 knockdown on IRE1 protein levels (Supplementary Fig. 6a–e) and on proliferation using an automated live-cell imaging instrument (Fig. 6a–e). G6374, but neither its epimer control G2642, nor the VHL ligand VH032, induced a concentration-dependent growth inhibition of OPM2 cells (Fig. 6a; Supplementary Fig. 6f), which require IRE1's enzymatic activity for optimal growth[37]; as well as AMO1 (Fig. 6b; Supplementary Fig. 6g) and KMS27 cells (Fig. 6c; Supplementary Fig. 6h), which require a nonenzymatic IRE1 scaffolding function[18]. In contrast, G6374 did not affect the proliferation rate of U2OS (Fig. 6d; Supplementary Fig. 6i) or Colo201 cells (Fig. 6e; Supplementary Fig. 6j), which do not have a significant dependency on IRE1[7,55]. These results show that VHL-based degradation of IRE1 selectively blocks the growth of IRE1-dependent cancer cell lines regardless of their mode of IRE1 dependency.

## Discussion

The discovery that certain cancer cell lines require IRE1 in a nonenzymatic scaffolding capacity, coupled with the failure of IRE1 kinase and/or RNase inhibitors to block growth in such settings[18,19], has created the need for a more comprehensive disruption strategy for IRE1. Although a CRBN-based degrader linked to an IRE1-RNase ligand has been reported[22,23], this compound afforded only partial degradation of IRE1, and its ability to inhibit cancer-cell proliferation was not determined. Our present work establishes an alternative strategy to degrade IRE1 by recruiting the CRL2 adapter VHL to a different site on IRE1, namely, the kinase domain. Furthermore, we report the cryoEM structure of a ternary complex between IRE1, a heterobifunctional degrader, and VHL. G6374 displayed significantly greater potency and induced much more complete degradation of endogenous IRE1 as compared to the previously reported CRBN-based IRE1 degrader. We performed a detailed biochemical characterization of the events that underlie G6374-induced IRE1 degradation. Our cryoEM and mutagenesis analyses shed important light on the interaction interfaces within the ternary complex at the atomic level. These insights should aid the future design of degraders for IRE1 and perhaps also for other serine/threonine kinase targets. Finally, our biological studies demonstrate that chemically-induced, VHL-based IRE1 degradation enables effective growth disruption of cancer cell lines that require IRE1 either enzymatically or nonenzymatically, while sparing IRE1-independent cells.

In designing an effective chemical degrader for IRE1, we leveraged the available knowledge regarding kinase-based IRE1 ligands[16], and ligands that can bind to VHL without abating CRL2[VHL] assembly[54,56]. We enabled IRE1 ubiquitination by linking the two ligands through a relatively flexible four-carbon alkyl linker of optimal length. G6374 induced an efficient and essentially complete degradation of endogenous IRE1 in cancer cell lines that express relatively high levels of the protein[18,37]. This work exemplifies an effective, VHL-based heterobifunctional degrader that targets an ER-localized transmembrane protein[20,57]. It is notable in this context that G6374-driven IRE1 degradation, which requires CRL2[VHL]-based ubiquitin transfer and proteasome activity, primarily proceeds independently of ERAD, which regulates the basal turnover of the IRE1 protein[58].

G6374 was capable of inducing IRE1 degradation even in the context of ER stress, demonstrating its ability to overcome dynamic changes in IRE1's phosphorylation and oligomerization during the UPR. Our cryoEM structural analysis suggests that G6374 captures the active conformation of the IRE1 kinase domain for degradation. In line with this, we observed comparable ubiquitination of nonphosphorylated and fully phosphorylated IRE1.KR protein. Furthermore, mutational disruption of the B2B dimer demonstrated that G6374 promotes comparable ubiquitination and degradation of both monomeric and dimeric IRE1 species. This finding further supports the conclusion that the compound can promote efficient IRE1 degradation

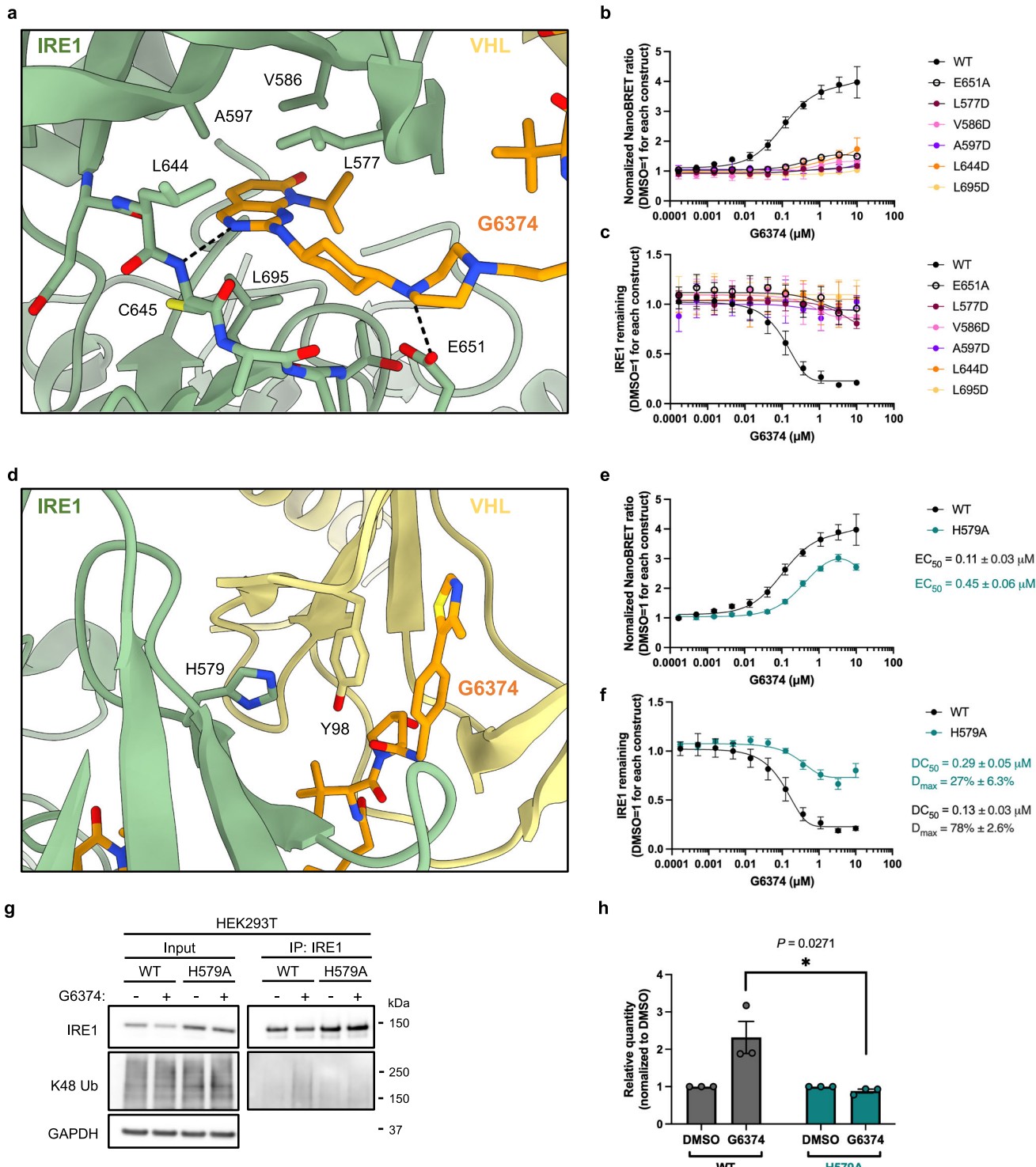

**Fig. 4 | Details of the VHL:G6374:IRE1 interface. a** A zoomed-in view of IRE1:G6374 interface at the ATP-binding pocket within the kinase domain of IRE1, with highlighted interactions. **b** Analysis of IRE1:VHL binding by NanoBRET for WT IRE1 versus mutants predicted to disrupt IRE1:G6374 interactions. **c** Analysis of IRE1 protein degradation by NanoBRET assay for WT IRE1 versus mutants predicted to disrupt IRE1:G6374 interaction. **d** A zoomed-in view of the IRE1:VHL interface with highlighted interactions. **e** Analysis of IRE1:VHL binding by NanoBRET for WT IRE1 versus the H579 mutant predicted to disrupt IRE1:VHL interaction. **f** Analysis of IRE1 protein degradation by NanoBRET assay for WT IRE1 versus H579 mutant predicted to disrupt IRE1:VHL interaction. **g** IP-IB analysis of IRE1 from HEK293T cells ectopically expressing IRE1 WT versus H579A, with or without G6374 (0.3 μM, 2 h), for K48-linked ubiquitination. **h** K48-linked ubiquitination signal from IRE1 IP in (**g**) normalized to their IRE1 IP signal. WT NanoBRET data were the same as in Fig. 2b, c. NanoBRET and IP-IB quantification were plotted with mean ± SEM from three biological replicates. Unpaired two-tailed Student's $t$-tests were used to assess differences between the mean ± SEM of the two groups.

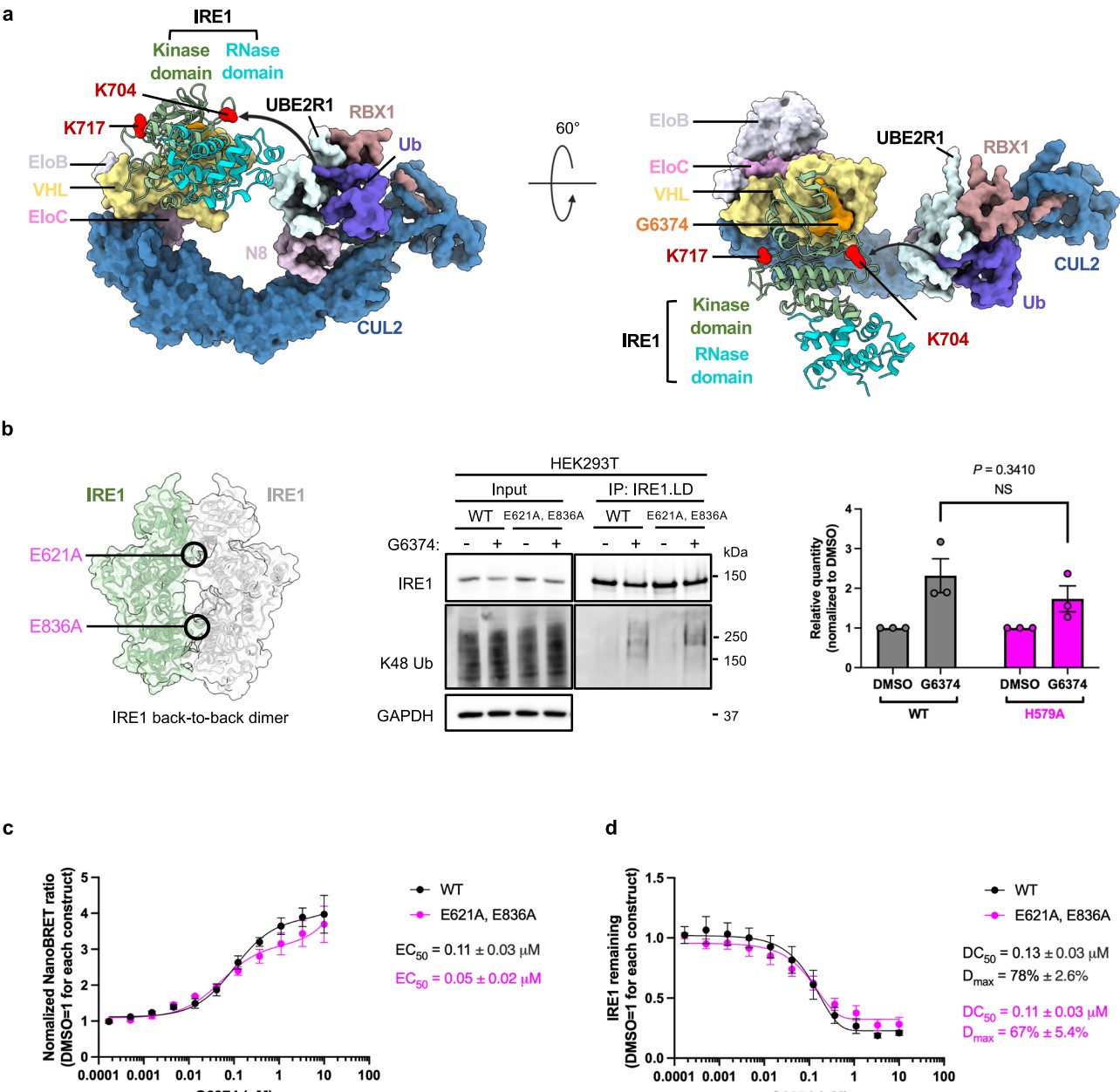

**Fig. 5 | G6374 can target monomeric IRE1 for degradation. a** Model of mono-meric IRE1 in complex with a full CRL2^VHL (PDB: 8RX0) assembly. The arrow illus-trates potential accessibility for Ub transfer by RBX1 to K704 on the bound IRE1 protomer (Note: we used the IRE1.KR structure from PDB: 6W3B to model the complex because the side chain of K717 from our structure is less well resolved). **b** Structure of the IRE1 B2B dimer, highlighting two mutations designed to disrupt dimerization (left). IP-IB analysis of WT versus E621A, E836A double-mutant IRE1, with or without G6374 (0.3 μM, 2 h) for K48-linked polyubiquitination (middle) with

K48-linked ubiquitination signal normalized to their IRE1 IP signal (right; WT data was the same as in Fig. 4h). Unpaired two-tailed Student's t-tests were used to assess differences between the mean ± SEM of the two groups. **c** Analysis of IRE1:VHL binding by NanoBRET for WT versus E621A, E836A double-mutant IRE1. **d** Analysis of IRE1 protein degradation by NanoBRET for WT versus E621A, E836A double-mutant IRE1. WT NanoBRET data were the same as in Fig. 2b, c. NanoBRET and IP-IB quantification were plotted with mean ± SEM from three biological replicates.

independent of the activation state. Whether an IRE1 ligand that shifts the kinase domain αC-helix into an inactive conformation would fur-ther augment or disrupt cellular IRE1 degradation has yet to be investigated.

The structural elucidation of the G6374-induced ternary complex enables assessment of the IRE1 B2B dimer in the context of a full CRL2^VHL assembly[54]. Based on structural alignment and modeling, additional lysine residues on IRE1.KR appear to be in sufficient proxi-mity to UBE2R1-Ub (Supplementary Fig. 7a, b) and could therefore be considered "primed" for ubiquitin transfer[54,56]. However, our empirical,

LC-MS/MS-based analysis identified with confidence only K704 and K717 as ubiquitinated. A plausible explanation for this apparent site restriction is that the addition of polyubiquitin chains at preferred lysine residues hinders modification at additional positions on IRE1. Consistent with this notion, mutation of K704 and K717 to arginine did not abolish ubiquitination on IRE1; rather, it altered the ubiquitination pattern as detected by Me-Ub conjugation.

Our structure-function experiments examined the impact of multiple point mutations on the interaction between G6374 and IRE1 as well as the association between IRE1 and VHL. This detailed analysis

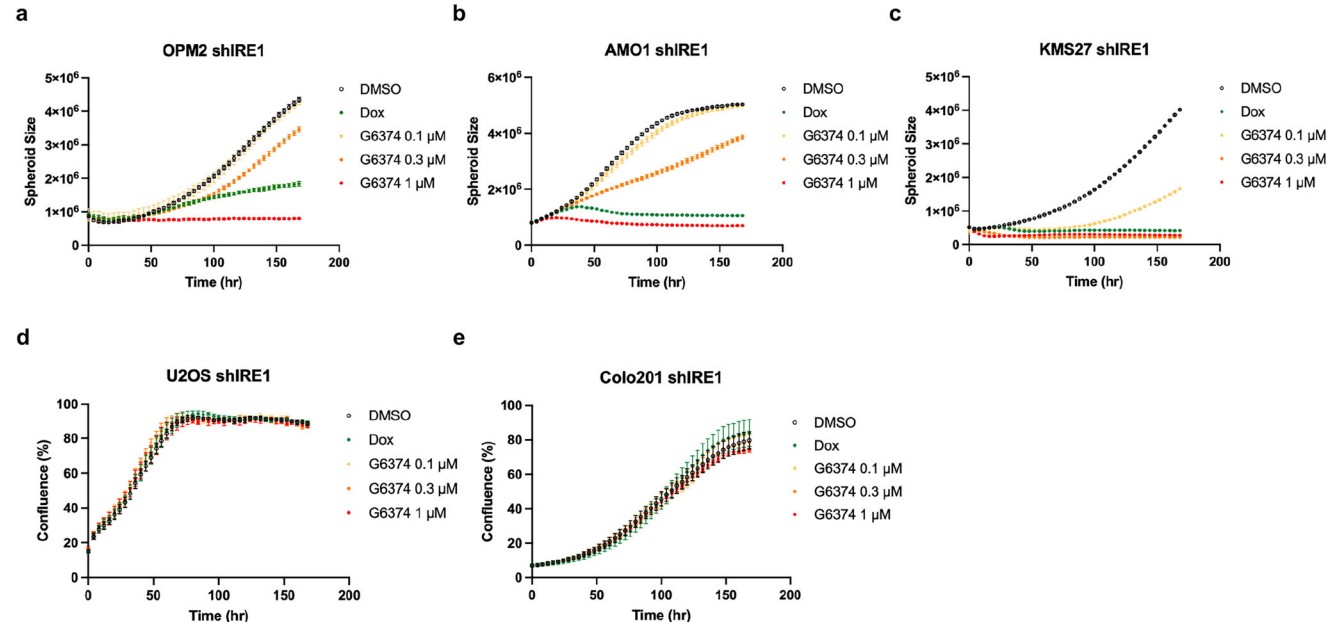

**Fig. 6 | G6374 selectively inhibits IRE1-dependent cancer-cell proliferation regardless of the underlying mode of dependency.** Incucyte data (plotted with at least three replicates, mean ± SD) for OPM2 (**a**), AMO1 (**b**), KMS27 (**c**), U2OS (**d**), or Colo201 (**e**) cells expressing Dox-inducible shRNA against IRE1, treated with a titration of G6374. Dox is used at 0.2 μg/mL as a control for IRE1 depletion.

provides specific insights that may prove to be important for designing even more potent and effective degraders for IRE1 and possibly for other kinases. Intriguingly, we detected an apparent weak intrinsic proclivity of IRE1 and VHL to interact. This invites future investigation of the biological significance behind such potential interaction, particularly given the known functional connection between IRE1 and HIF-1α—a major physiological substrate of VHL in the context of hypoxia, itself linked to ER stress[59–61]. Furthermore, the weak potential interaction between VHL and IRE1 raises the tantalizing possibility of finding a molecular glue that may promote the ubiquitination of IRE1, as suggested by Zheng and coworkers[62].

G6374 has the same VHL ligand moiety as two previously described heterobifunctional degraders of the BRD4 protein, namely, MZ1[51] and CIP-1[50], as well as degraders of the WDR5 protein, i.e., MS33 and MS67[63]. We compared the surface exposure of VHL in the BRD4:MZ1:VHL complex, BRD4:CIP-1:VHL complex, and IRE1:G6374:VHL complex (Supplementary Fig. 8). Notably, despite the shared VHL binding motif and the relative flexibility of the linkers, each compound leads to a different interface being utilized by VHL for productive protein-protein interaction with the neosubstrate, underscoring the plasticity and productivity of ubiquitin ligases to modify their targets. Furthermore, the conjugation of our IRE1 ligand to a VHL binding motif led to an increase in degradation selectivity. This improvement in functional selectivity is an emerging theme in the heterobifunctional degrader field, highlighting the importance of the protein-protein interaction between VHL and the recruited target[51].

Increasingly, structural evidence has begun to emerge that rather than recruiting neosubstrates in a "beads-on-a-string" manner, PRO-TACs help to bring proteins into close proximity to ubiquitin ligases and then cooperatively promote productive protein-protein interactions between the E3 and its neosubstrate. Although most interfaces are limited in scope, structure-based PROTAC optimization has led to improvements in total BSA between ligase and target to promote degradation[50,51,57,63,64] (Supplementary Table 3). G6374 promotes a comparable BSA between VHL and IRE1 as some highly potent VHL-based PROTACs that target other substrates. We anticipate that our

structure will help guide the design of improved, rigidified linkers that promote a tighter, more intimate interaction between VHL and IRE1.

Gratifyingly, our cellular studies showed that G6374 induces strong growth inhibition in cancer cell lines previously shown to depend on IRE1 either enzymatically or nonenzymatically. On the other hand, G6374 did not significantly disrupt the growth or viability of two cancer cell lines that do not rely on IRE1 for growth. While this selectivity is reassuring, in vivo studies will be needed to ascertain the pharmacokinetics, pharmacodynamics, efficacy, and tolerability of G6374 or its future derivatives.

In summary, we have developed a unique molecular strategy to degrade endogenous IRE1 by building on prior knowledge regarding ligands of the IRE1 kinase, VHL, and the characterization of other heterobifunctional degraders. Our efforts to date have culminated in a potent, effective, and selective compound that drives complete IRE1 degradation and specifically inhibits the growth of IRE1-dependent cancer cells regardless of their underlying type of IRE1 requirement. This advance is significant, given that more than half of MM cell lines that display IRE1 dependency require nonenzymatic IRE1 function[18,19]. Our study provides a proof-of-concept that VHL can be recruited to efficiently degrade an ER-transmembrane protein. A fascinating question emerges: how is the ubiquitinated IRE1 protein delivered from the ER membrane to the proteasome? Finally, our observations advance current knowledge toward a more effective therapeutic disruption of IRE1's pathological roles in cancer and perhaps other diseases.

## Methods
### High-throughput PROTAC screening
The discovery of G6374 was achieved using high-throughput screening of HEK293T cells expressing HiBiT-tagged human IRE1α, followed by targeted Structure Activity Relationship analysis using a Meso Scale Discovery (MSD) assay of endogenous IRE1 in the RPMI-8226 cell line. Cells were plated on Day 1 in growth medium and allowed to recover overnight at 37 °C before compounds treatment on Day 2. Well contents were removed, and cells were lysed at ambient temperature with 20 μL/well of 1X RIPA buffer containing 50 mM Tris, 150 mM NaCl,

2 mM EDTA, 1% NP-40, 0.1% SDS, and 1X Phosphatase and Protease Inhibitor Cocktail (Invitrogen, #78440) on Day 3. Cell lysates were assayed fresh or frozen at −80 °C for assay on a later date. For assay of fresh lysates, an MSD GAR (Goat Anti-Rabbit coated) single-spot 384-well plate was blocked for 1 h on Day 2, then washed, and a rabbit anti-IRE1 RNase-domain capture antibody was added to all wells and incubated at 4 °C overnight. On Day 3, the capture antibody was washed off, and 10 μL of cell lysates were added to the MSD plates. The lysate incubation and all subsequent incubations for a mouse anti-human IRE1 lumenal-domain detection antibody (made in-house at Genentech, described in Shemorry et al.[65]) and an anti-mouse SulfoTag labeling antibody were carried out at ambient temperature with agitation for 2 h each. Plates were washed for a final time after the SulfoTag labeling, then Read Buffer T (MSD, # R92TC-1) was added, and plates were read immediately on a MESO SECTOR S 600 imager (MSD). All MSD plate washes were done on a BlueWasher (Blue Cat Bio) or BioTek EL405 washer (Agilent). All antibodies were diluted in MSD Antibody Dilution Buffer containing 50 mM Tris, pH 7.5, 150 mM NaCl, 0.02% Tween 20, and 1% BSA. IRE1-dependent electrochemiluminescence values were normalized to minimum and maximum effect controls, and the normalized data were fit to a 4-parameter sigmoidal equation to determine degradation for each compound.

### Chemical synthesis

The synthesis and characterization of G6374 are detailed in the Supplementary Information.

### Cell lines

The following cell lines were used in this study: AMO1, KMS27, OPM2, U2OS, Colo201, H929, HCT116, and HEK293T. All cell lines were authenticated by DNA sequencing and tested negative for mycoplasma contamination.

### Immunoblotting (IB)

Cells were lysed in buffer containing 20 mM HEPES 7.2, 150 mM NaCl, and 1% Triton X-100, supplemented with fresh protease and phosphatase inhibitors (Invitrogen, #78440), cleared by centrifugation at $12,000 \times g$ for 10 min, and analyzed by BCA protein assay (Thermo Fisher Scientific, #23227). Equal protein amounts were loaded, separated by SDS-PAGE, electrotransferred to nitrocellulose membranes using the iBLOT2 system (Invitrogen), and blocked in 5% nonfat milk solution for 60 min. Membranes were probed with primary antibodies at 1:1000 dilution in 5% milk at 4 °C overnight. Signal was detected using appropriate horseradish peroxidase (HRP)-conjugated secondary antibodies at 1:10,000 in 5% milk at room temperature for 1–2 h. Bands were visualized by the Azure 600 western blot imaging system (Azure Biosystems). For in vitro ubiquitination assays performed with Me-Ub, bands were visualized by the Odyssey CLx Imager (LI-COR) after incubating the blots with the IRDye 800CW (LI-COR) secondary antibodies at 1:10,000 in Intercept Blocking Buffer (LI-COR) at room temperature for 1–2 h.

Primary antibodies used in the studies: IRE1α (14C10) Rabbit mAb (Cell Signaling, #3294), GAPDH (14C10) Rabbit mAb (Cell Signaling, #2118), CDK7 Antibody (Cell Signaling, #2090), CDK9 (C12F7) Rabbit mAb (Cell Signaling, #2316), GSK-3α (D80E6) Rabbit mAb (Cell Signaling, #4337), CSF-1R/M-CSF-R Antibody (Cell Signaling, #3152), SAPK/JNK Antibody (Cell Signaling, #9252), JNK2 (56G8) Rabbit mAb (Cell Signaling, #9258), JNK3 (55A8) Rabbit mAb (Cell Signaling, #2305), MELK Antibody (Cell Signaling, #2274), PKD/PKCμ (D4J1N) Rabbit mAb (Cell Signaling, #90039), TAO2 antibody (Novus Biologicals, NBP2-20563), K48-linkage Specific Polyubiquitin Antibody (Cell Signaling, # 4289), VHL Antibody (Cell Signaling, #68547), LC3A/B (D3U4C) XP® Rabbit mAb (Cell Signaling, #12741), p-IRE1 antibody (made in-house at Genentech, described in Chang et al.[13]), K11 linkage-specific antibody (made in-house at Genentech, described in Matsumoto et al.[66]).

Secondary antibodies used in the studies: Peroxidase AffiniPure™ Donkey Anti-Rabbit IgG (H+L) (Jackson Immuno Research Laboratories, #711-035-152), Peroxidase-AffiniPure Goat Anti-Mouse IgG (H+L) (Jackson Immuno Research Laboratories, #115-035-003), Goat Anti-Mouse IgG2a Human ads-HRP (SouthernBiotech, #1080-05), IRDye® 800CW Goat Anti-Rabbit IgG (LI-COR, #926-32211).

### Immunoprecipitation (IP)

**IP of cellular proteins.** IRE1.LD antibodies were conjugated to agarose beads using the AminoLink® Plus Immobilization Kit (Thermo Fisher, #44894). Cell pellets were lysed on ice for 20 min in lysis buffer containing 20 mM Tris 7.5, 150 mM NaCl, 1% Triton X, and 1 tablet per 50 ml buffer of protease inhibitor cocktail (Roche, #11836170001). 10 mM NEM was added to stabilize the IB signal for K48-Ub. Cell lysates were cleared by centrifugation at $16,000 \times g$ at 4 °C. For IP of endogenous IRE1, 20 μl of anti-IRE1.LD-conjugated beads were added to ~3 mg of total protein and rotated at 4 °C overnight. For IP of over-expressed IRE1 from HEK293T cells, 20 μL of anti-IRE1.LD-conjugated beads were added to ~0.4 mg/mL of total protein lysate and rotated at 4 °C overnight. Beads were then harvested by centrifugation at $500 \times g$ at 4 °C for 2 min, washed with lysis buffer three times, then quenched in LDS sample buffer (Invitrogen, NP0007) containing reducing agent (Invitrogen, NP0009), and boiled at 70 °C for 10 min.

**IP of recombinant proteins.** Fifty nanomolars of recombinant IRE1.KR, 50 nM of recombinant VCB, and 100 nM of G6374 or an equal volume of DMSO were mixed and incubated in binding buffer containing 20 mM HEPES, 7.2, 150 mM NaCl, 2 mM Glutamate, 2 mM MgCl₂, and 10% glycerol at room temperature for 1.5 h. One microliter of IRE1 antibody (Cell Signaling, #3294) was added per sample, followed by incubation at 4 °C overnight. Twenty microliters of protein A/G agarose beads (Pierce, #20421) were added per sample, followed by rotation at room temperature for 2 h. Beads were then washed four times with buffer containing 20 mM HEPES 7.2, 300 mM NaCl, 10% glycerol, and 1% Triton X-100, resuspended in LDS sample buffer (Invitrogen, NP0007) containing reducing agent (Invitrogen, NP0009), and boiled at 70 °C for 10 min.

### Kinase selectivity analysis

The kinome selectivity of G6374 was assessed at 1 μM for 1 h versus a panel of 220 human kinases (SelectScreen Kinase Profiling Services, Thermo Fisher Scientific). The assays were conducted using either the Z'-LYTE technology to assess inhibition of enzyme activity ([ATP] = Km) or the TR-FRET-based LanthaScreen Eu Kinase Binding assay to assess competitive displacement of an ATP site binding probe. Assay results were normalized to positive and negative controls to determine percent inhibition. Image was created by KNIME Report Designer (KNIME.com AG, Zurich, Switzerland). Kinases for which ATP binding was inhibited by more than 90% were further examined by IB analysis for depletion by G6374 in AMO1 cells (4 h incubation).

### siRNA knockdown

siRNA knockdown was performed through reverse transfection in a Falcon tube. For each siRNA, 15 μL of 50 μM siRNA and 75 μL RNAiMAX Transfection Reagent (Invitrogen, #13778150) were gently mixed in 2.5 mL Opti-MEM (Gibco, #31985062), incubated at RT for 10 min, then added to the Falcon tube containing $1.25 \times 10^7$ cells suspended in 37.5 mL media. The transfected cell suspension was distributed into 6-well plates (3 mL/well) to ensure uniform knockdown across wells. G6374 treatments were performed at 1 μM using a reverse time course design, in which treatments were initiated at different intervals prior to the endpoint, allowing all samples to be harvested simultaneously after 48 h of siRNA knockdown.

**Table 1 | siRNAs used for VHL knockdown**

| Reference number for each siRNA | Target sequence |
|---|---|
| J-003936-09 | AGGCAGGCGUCGAAGAGUA |
| J-003936-10 | CCACCCAAAUGUGCAGAAA |
| J-003936-11 | GGAGCGCAUUGCACAUCAA |
| J-003936-12 | CCAAUGGAUUCAUGGAGUA |

The VHL knockdown was performed using the following siRNAs provided as a mixed pool by Dharmacon (Cat # L-003936-00-0005) (Table 1).

**Crosslinking assay**

Cells were lysed in PBS supplemented with 1% Triton X-100 and fresh protease and phosphatase inhibitors (Invitrogen, #78440), incubated on ice for 10 min, and cleared by centrifugation at $12,000 \times g$ for 10 min. DSS (Thermo Fisher Scientific) was added to the lysate at 250 μM final concentration, followed by incubation at RT for 1 h. The reaction was quenched using a 1 M Tris pH 7.5 solution to a final concentration of 100 mM for 15 min at RT. Protein concentration was determined by BCA assay. Thirty micrograms of total protein per sample were loaded onto the SDS-PAGE gel for IB analysis.

**NanoBRET assay**

$8 \times 10^5$ 293T cells per well were seeded in 6-well plates (Costar, #3516) and incubated at 37 °C, 5% $CO_2$ for 6 h. A mixture of 0.2 μg ERN1 (IRE1)-NanoLuc Fusion Vector (Promega, NV1321), 2 μg HaloTag-VHL Fusion Vector (Promega, N2731), 6 μl FuGENE HD Transfection Reagent (Promega, E2311), and 100 μL Opti-MEM I reduced serum medium (Gibco, #31985062) per well was incubated at room temperature for 20 min before adding to cells. For expression of IRE1 mutants, site-directed mutagenesis was performed using the ERN1-NanoLuc Fusion Vector as the DNA template, and the transfection mixture was prepared in the same way as WT IRE1. After 18 h, cells from 6-well plates were trypsinized, resuspended in Opti-MEM I reduced serum medium, and reseeded into 96-well plates (Costar, #3917), with or without the addition of HaloTag NanoBRET 618 Ligand (Promega, G9801). After 18 h of incubation, a titration of G6374 (dilutions made in Opti-MEM I reduced serum medium) was added to the cells. After 2 h of G6374 treatment, NanoBRET Nano-Glo Substrate (Promega, N1571) was added to cells. The 96-well plates were imaged by GloMax Discover Microplate Reader (Promega, GM3000) with the default "BRET: NanoBRET 618" protocol (Donor: 450 BP; Acceptor: 600 LP; Integration: 0.3 s). Data analysis was performed according to the user manual.

**Protein purification**

Recombinant human IRE1.KR (G547-L977) was purified using an established protocol[25]. Briefly, the N-terminal His6-tagged IRE1.KR (G547-L977) with a TEV protease cleavage site was expressed in Sf9 cells from an intracellular BEVS expression vector. Cell pellet was resuspended in lysis buffer containing 50 mM HEPES pH 8.0, 300 mM NaCl, 10% glycerol, 1 mM $MgCl_2$, 1:1000 benzonase, EDTA-free PI tablets (Roche), 1 mM TCEP, and 5 mM imidazole. The sample was lysed by sonication, clarified by centrifugation at $15,000 \times g$ for 45 min, then affinity-purified using packed Ni-NTA Superflow beads (Qiagen). The His6-tag was removed by TEV protease made in-house. The protein solution contained a mixture of phosphorylated and unphosphorylated IRE1. To separate the 2 pools, protein solution was loaded on a 5 mL pre-packed Q-HP column (Cytiva) and eluted with a very shallow gradient (50–300 mM NaCl over 70 CV). Fully phosphorylated fraction was collected separately, while the rest of the protein fractions were pooled and incubated with Lambda phosphatase for 2 h at room temperature. Each pool was then purified by size-exclusion chromatography (Superdex 200 Increase 10/300 GL, Cytiva). Protein was stored in buffer containing 50 mM HEPES 7.5, 200 mM NaCl, 10% glycerol, and 1 mM TCEP. Fully phosphorylated (3P) and dephosphorylated (0P) IRE1.KR were confirmed by LC-MS.

The N-terminal His6-tagged VHL (E55-D213) was co-expressed with EloB (M1-Q118) and EloC (M17-C112) in BL21-Gold (DE3) cells (Agilent). Cells were resuspended in buffer containing 20 mM Tris, pH 7.5, 150 mM NaCl, 10% glycerol, 1 mM TCEP, and 3 mM imidazole, lysed by microfluidizer (Hyland Scientific), and clarified by centrifugation. The clarified supernatant was affinity-purified using Ni-NTA Superflow beads (Qiagen), followed by ion-exchange chromatography (Mono Q 10/100 GL, Cytiva) and size-exclusion chromatography (Superdex 200 Increase 10/300 GL, Cytiva). Protein was stored in buffer containing 20 mM Tris, pH 7.5, 150 mM NaCl, and 0.5 mM TCEP.

CUL2:RBX1 complex was expressed on a polycistronic vector containing a TEV-cleavable N-terminal His6-GST tag on RBX1 in Sf9 insect cells. Cells were lysed in buffer containing 50 mM HEPES, pH 8.0, 250 mM NaCl, 0.5% Triton X-100, and 1 mM TCEP. Lysates were clarified by centrifugation at $31,340 \times g$ for 1 h. The supernatant was then incubated with Ni-NTA agarose resin at 4 °C for 1 h. The Ni-NTA resins were then washed with lysis buffer supplemented with 30 mM imidazole, followed by protein elution in lysis buffer supplemented with 300 mM imidazole, and TEV cleavage in dialysis buffer overnight (20 mM Tris, pH 8.0, 300 mM NaCl, 5% Glycerol, and 1 mM TCEP), reverse Ni-NTA purification, and finally size-exclusion chromatography using a 26/600 Superdex 200 column (Cytiva) equilibrated in 20 mM HEPES, pH 8.0, 200 mM NaCl, 5% glycerol, and 1 mM TCEP.

Neddylation E1 and E2 enzymes (NAE1:UBA3 and UBE2M) and Ubiquitin E1 and E2 enzymes (UBE1 and UBE2R1) were expressed in BL21-Gold (DE3) cells. Cells were lysed in 50 mM Tris pH 7.5, 500 mM NaCl, 10% glycerol, and 1 mM TCEP. Proteins were purified by affinity chromatography through an N-terminal His6-tag using Ni-NTA agarose, followed by steps similar to CUL2:RBX1 protein purification. The eluent from Ni-NTA beads was concentrated and run over a 16/600 S200 column equilibrated in 20 mM Tris, pH 7.5, 200 mM NaCl, 5% glycerol, and 1 mM TCEP.

**In vitro ubiquitination assay**

The protocol for the in vitro ubiquitination experiment was adapted from a previous publication[38].

The ubiquitination reaction buffer contained 30 mM Tris pH 7.5, 100 mM NaCl, 5 mM $MgCl_2$, 2 mM ATP, and 2 mM DTT. Recombinant proteins and G6374 with high stock concentrations were pre-diluted in binding buffer containing 30 mM Tris, pH 7.5, 100 mM NaCl, 1 mM DTT for the purpose of accurate pipetting.

The 0.2 μM IRE1: 0.2 μM G6374 (or DMSO) complex was pre-incubated for 15 min at room temperature, then incubated with the neddylation reaction mixture containing 0.2 μM VHL:EloB:EloC, 0.2 μM CUL2-RBX1, 0.1 μM NAE1:UBA3, 1 μM UBE2M, 3 μM NEDD8 for another 15 min at room temperature. Meanwhile, the ubiquitination reaction mixture containing 1 μM UBE1, 5 μM UBE2R1 (or UBE2D1, UBE2D2), and 500 μM ubiquitin or methyl-ubiquitin was prepared and pre-incubated for 15 min before incubation with the neddylation reaction mixture containing IRE1:G6374. Samples were taken at each timepoint and quenched in NuPage LDS buffer supplemented with NuPAGE sample reducing agent, then boiled at 95 °C for 10 min. Ubiquitinated products were visualized by IB with the IRE1 antibody.

UBE2D1 (E2-616-100), UBE2D2 (E2-622-100), ubiquitin (U-100H-10M), methylated ubiquitin (U-502-01M), and NEDD8 (UL-812-500) were purchased from Biotechne.

**Detection of polyubiquitin linkage and site of ubiquitination by mass-spectrometry**

For the gel with 500 μM ubiquitin, we excised and divided the region between 100 kD and >250 kD. For 500 μM methyl-ubiquitin, we did the same for the region between 50 kD and 100 kD. The gel slices were

destained with 50 mM ammonium bicarbonate/50% acetonitrile, reduced with 50 mM Dithiothreitol/50 mM ammonium bicarbonate for 30 min at 37 °C, followed by alkylation with 50 mM iodoacetamide/ 50 mM ammonium bicarbonate, and digested with 0.02 µg/µl of trypsin in 50 mM ammonium bicarbonate overnight at 37 °C. The digested slices' supernatant was dried and desalted using C18 stage tips as previously described[67].

LC-MS/MS analysis was performed by injecting 2 µL of 10 µL of each gel slice on an Orbitrap Lumos mass spectrometer (Thermo Fisher Scientific) coupled to a Dionex Ultimate 3000 RSLC (Thermo Fisher Scientific) employing a 25 cm IonOpticks Aurora series column (IonOpticks) with a gradient of 2–30% Buffer B (98% ACN; 2% H2O with 0.1% FA; flow rate, 300 nLeaving/min). Global proteome samples were analyzed with a total run time of 69 min. The samples on Orbitrap Eclipse were collected with FTMS1 scans at 240,000 resolution with an AGC target of 250% and a maximum injection time of 50 ms. FTMS2 scans on precursors with charge states of 2–6 were collected at 15,000 resolution with collision-induced dissociation fragmentation at a normalized collision energy of 35%, an AGC target of 400%, and a max injection time of 100 ms.

The data were searched using Comet v2019.01 with the following parameters: UniProt human database August 2021 version, including 218,136 Swiss-Prot sequences of canonical and protein isoforms, plus common contaminants and decoys; static modifications including Cys carbamidomethylation (+57.0215); and variable modifications including Met oxidation (+15.9949) and Lys (+114.042927). The peptide false discovery rate was filtered to <1% using a linear discriminant algorithm.

## TMTPro sample preparation

One milligram of protein lysate in denaturing buffer (8 M Urea, 20 mM HEPES, pH 8.0) was reduced (5 mM dithiothreitol (DTT), 45 min at 37 °C), alkylated (15 mM iodoacetamide (IAA), 20 min at room temperature in the dark), and quenched (5 mM DTT, 15 min at room temperature in the dark). Proteins were pelleted by chloroform-methanol precipitation. The resulting pellet was resuspended in denaturing buffer, diluted to 4 M Urea, and digested for 4 h at 37 °C with lysyl-endopeptidase (Wako) at an enzyme to protein ratio of 1:100. The sample was further diluted to 1.3 M Urea and subjected to overnight enzymatic digestion at 37 °C with sequencing grade trypsin (Promega, enzyme: protein ratio = 1:50). Resultant peptides were acidified with 20% Trifluoroacetic acid (TFA, 1% final concentration), centrifuged at 18,000 × g for 15 min, and desalted using a Sep-Pak C18 column (Waters).

For global proteome samples, 100 µg of peptides from each sample was dissolved in 100 mM HEPES, pH 8.0 (1 mg/mL). Isobaric labeling was performed using TMTPro18-plex reagents (Thermo Fisher). Each unit (0.5 mg) of TMT reagent was allowed to reach room temperature immediately before use, spun down on a benchtop centrifuge, and resuspended with occasional vortexing in 20 µL anhydrous acetonitrile (ACN) prior to mixing with peptides (18% final ACN concentration). After incubation at room temperature for 1 h, the reaction was quenched for 15 min with 20 µL of 5% hydroxylamine. Labeled peptides were combined in equimolar ratios and dried. The TMTpro-labeled sample was re-dissolved in 80 µL 0.1% TFA, centrifuged at 16,000 × g, and the supernatant was processed further. Offline high pH reversed-phase fractionation was performed on a 1100 HPLC system (Agilent) using an ammonium formate-based buffer system. Peptides (400 µg) were loaded onto a 2.1 × 150 mm 3.5 µm 300 Extend-C18 Zorbax column (Agilent) and separated over a 75-min gradient from 5% to 85% ACN into 96 fractions (flow rate = 200 µL/ min). The fractions were concatenated into 24 fractions, mixing different parts of the gradient to produce samples that would be orthogonal to downstream low pH reversed phase LC-MS/MS. Fractions were dried and desalted using C18 stage tips as previously described[67].

Peptides were lyophilized and resuspended in 10 µL Buffer A (2% ACN, 0.1% formic acid) for LC-MS/MS analysis.

## Mass spectrometry

For the global proteome, LC-MS/MS analysis was performed by injecting 1 µL of each fraction on an Orbitrap Eclipse mass spectrometer (Thermo Fisher) coupled to a Dionex Ultimate 3000 RSLC (Thermo Fisher) employing a 25 cm IonOpticks Aurora Series column (IonOpticks, Parkville, Australia) with a gradient of 2% to 30% buffer B (98% ACN, 2% H2O with 0.1% FA, flow rate = 300 nL/min). Global proteome samples were analyzed with a total run time of 95 min. The Orbitrap Eclipse with FAIMS Pro DUO of −40, −60 CV collected FTMS1 scans at 120,000 resolution with an AGC target of $1 \times 10^6$ and a maximum injection time of 50 ms. FTMS2 scans on precursors with charge states of 3–6 were collected at 15,000 resolution with CID fragmentation at a normalized collision energy of 30%, an AGC target of $2 \times 10^4$, and a max injection time of 100 ms.

Real-time database search (RTS) was performed prior to acquisition of MS3 spectra using ThermoRTS. The following RTS parameters were used for global proteome analysis: Uniprot human database August 2021 version, including 218,136 Swiss-Prot sequences of canonical and protein isoforms, plus common contaminants and decoys; static modifications included Cys carbamidomethylation (+57.0215), Lys and n-term TMTPro (+304.207146); variable modifications included Met oxidation (+15.9949) and Tyr TMTPro (+304.207146). Offline search was performed using Comet v.2019.01 with parameters matched to the RTS search. Peptide FDR was filtered to <1% using the Linear Discriminator Algorithm. TMT reporter ions produced by the TMT tags were quantified with the Mojave in-house software package by calculating the highest peak within 20 ppm of theoretical reporter mass windows and correcting for isotope purities.

Quantification and statistical testing of global proteome proteomics data were performed by the MSstatsTMT_2.0.1 R package[68]. Multiple fractions from the same TMT mixture were combined in MSstatsTMT v2.0.1[68]. In particular, if the same peptide ion was identified in multiple fractions, only the single fraction with the highest maximal reporter ion intensity was kept. Global median normalization was carried out to reduce the systematic bias between channels.

## Fluorescence polarization

Assays were performed in duplicates on ProxiPlate-384 F Plus (Revvity, #6008260) with a total well volume of 16 µL. Each well contained 50 nM of FAM-labeled HIF-1α peptide (FAM-DEALA-Hyp-YIPD, biotechne, #7287), 500 nM of VCB protein, and a titration of G6374 (2-fold serial dilutions from 4 µM to 625 nM, then 1.3-fold serial dilutions to 284.5 nM, followed by 2-fold serial dilutions to 8.9 nM) or IRE1.KR:G6374 (constant concentration of 2.4 µM IRE1.KR and the same serial dilution of G6374 as mentioned above) in 50 mM HEPES pH 7.2, 100 mM NaCl, 1 mM TCEP, and 0.002% Tween 20 with a final concentration of DMSO of 1.5%. The plate was incubated at room temperature in the dark, first for 5 min on a shaker at 300 rpm, then for another 30 min on the bench. FP was measured using the EnVision Multimode Plate Reader (PerkinElmer) with fluorescence excitation and emission wavelengths of 480 nm and 535 nm. The percentage of HIF-1α displacement was calculated by normalization to maximum fluorescence anisotropy (or 0% displacement of the HIF-1α peptide) and minimum fluorescence anisotropy (or 100% displacement of the HIF-1α peptide), which were determined by "top" and "bottom" values of the curve fit of fluorescence anisotropy against G6374 concentration in GraphPad Prism 10 (GraphPad Software, Inc.). The $IC_{50}$ values were then determined by curve fitting with the percentage of HIF-1α displacement against G6374 concentration in GraphPad Prism 10 (GraphPad Software, Inc.).

## Surface plasmon resonance

All SPR experiments in this paper were recorded on a Biacore S200 instrument; all protein captures and data collection were at 20 °C.

For binary interactions with G6374: a Xantec SAHC 200 M (streptavidin) chip was inserted into the instrument and primed into running buffer [50 mM HEPES (pH 7.4), 150 mM NaCl, 0.001% Tween 20, 0.2% PEG3350, 0.25 mM TCEP, and 2% DMSO]. Biotinylated human IRE1.KR protein was captured on FC2 to 2-4000 RU, biotinylated VHL in complex with Elongin B (VHL/EloB) was captured on FC4 to 1-2000 RU, data were collected, and all referenced to FC1. Affinity for G6374 was measured using a titration series, injected from low to high with an 8-point, 2-fold dilution dose response with a top concentration of 0.5 µM in both single cycle (single cycle kinetics (SCK)) and multi cycle kinetics (MCK) formats. For MCK experiments, an association time of 45 s and a dissociation time of 200 s with a flow rate of 50 µL/min. For SCK experiments, an association time of 45 s and a dissociation time of 2000 s with a flow rate of 50 µL/min were used. All data were analyzed using Insight software (Cytiva) using a 1 to 1 kinetic binding model.

For Binary protein interaction between IRE1 and VHL: a 2D Xantec SAP (streptavidin) chip was inserted into the instrument and primed into running buffer [50 mM HEPES (pH 7.4), 150 mM NaCl, 0.01% Tween 20, 0.2% PEG3350, 0.25 mM TCEP, 100 µg/ml ovalbumin and 2% DMSO]. Biotinylated human IRE1.KR protein was captured on FC2 to 1-200 RU; data were collected and referenced to FC1. Binary affinity with VHL was measured using a titration series, injected from low to high with an 8-point, 2-fold dilution dose response with a top concentration of 5 µM in SCK formats. Ternary complex affinity was measured using a fixed concentration of 0.5 µM his-VHL/EloB pre-mixed with a titration series of G6374 injected from low to high with an 8-point, 2-fold dilution dose response with a top concentration of 0.125 µM in SCK format. Ternary complex titration was blanked to 0.5 µM VHL/EloB without compound.

SCK experiments were run with an association time of 90 s and a dissociation time of 2000 s, with a flow rate of 30 µL/min. All data were analyzed using Insight software (Cytiva) using a 1 to 1 kinetic binding model.

## Purification of the recombinant IRE1:G6374:VCB complex

Recombinant IRE1, VHL/EloB/EloC, and G6374 were mixed at 1:1:1 in a final volume of 450 µL of buffer, each with a final concentration of 12 µM. The complex was incubated at 4 °C overnight (16 h) and then centrifuged at 4 °C at 16,000 × g for 2 min before injection to SEC (Superdex 200 Increase 10/300 GL, Cytiva). The SEC buffer contains 20 mM HEPES 7.2, 150 mM NaCl, and 1 mM DTT. Fractions that correspond to the complex were confirmed by Coomassie staining on a 4–12% Bis-Tris SDS-PAGE gel, collected and concentrated to ~1 mg/mL by a 10 kDa centrifugal filter (Millipore, UFC80100), then flash-frozen in liquid nitrogen and stored at −80 °C.

## CryoEM grid preparation

0.2 mg/mL of the IRE1:G6374:VCB complex purified from SEC was crosslinked with 500 µM BS3 (Thermo Scientific, A39266) at room temperature for 60 min in buffer containing 20 mM HEPES 7.2 and 150 mM NaCl. BS3 was then quenched with 90 mM Tris 7.5 for 10 min. CTAB was added to a final concentration of 0.005% immediately before grid preparation. Our initial attempts at cryoEM grid preparation, which had not included pre-treatment with BS3 and the addition of CTAB, had led to unstable complexes on the grid, in addition to biased orientations of the particles. BS3 crosslinking helped with keeping the complexes intact, while CTAB circumvented the preferred orientation issue.

Prior to sample application, a holey gold grid (UltrAuFoil 25 nm R1.2/1.3 300 mesh; Quantifoil, Großlöbichau) was coated with a hydrophilic self-assembled PEG monolayer to improve particle distribution[69]. Each grid was then rinsed in isopropanol and dried completely before 3 µL of the complex was pipetted onto the grid, which was then blotted and plunge frozen in liquid ethane using a Vitrobot Mk IV (Thermo Fisher Scientific), at a temperature of 4 °C, 100% relative humidity, a blot force of 7, and a 2 s blot time.

The first electron microscopy dataset was collected with Glacios equipped with a Gatan K2 Summit direct detector (Thermo Fisher Scientific). 978 movies were recorded at an operating voltage of 200 kV, 36,000x nominal magnification (calibrated pixel size 1.148 Å/px) and a total electron exposure of ~47 e/Å². The second electron microscopy data were collected with a Titan Krios G2 equipped with a Falcon4i direct electron detector, SelectrisX energy filter, and EPU automation software (Thermo Fisher Scientific). In total, 12,400 movies were recorded at an operating voltage of 300 kV, 165,000× nominal magnification (calibrated pixel size 0.731 Å/px), energy filter slit width 20 eV, defocus range −0.6 to −3.5 µm, and a total electron exposure of ~45 e/Å².

## CryoEM data processing

CryoSPARC (version 4.6) was used for image processing. CryoSPARC Live was used for movie motion correction, CTF estimation. In total, 5,101,645 particles were picked based on an ab initio 3D model generated from a lower magnification screening dataset collected from the same grid. Subsequent 2D classification and heterogeneous refinement resulted in a final class of 400,055 particles. After reference-based motion correction, non-uniform refinement with CTF refinement (per-particle defocus, global beam tilt, and trefoil) followed by map denoising[70] produced a map with a global resolution of ~2.6 Å (see Supplementary Fig. 3b for data processing workflow).

Ligand restraint for G6374 was generated by Grade[71] (version 1.2.20). The model was built by rigid fitting of crystal structures of 6W3B and 5T35 into the cryoEM map using ChimeraX (version 1.7.1), followed by real-space refinement in COOT (version 0.9.6) and refinement in Phenix (version 1.21). Structure-related figures were made with ChimeraX (version 1.7.1) and PyMOL (version 2.5.2).

## Statistics and reproducibility

All statistical analyses were performed using GraphPad Prism 10 (GraphPad Software, Inc.). Unpaired two-tailed Student's $t$-tests were used to assess differences between the mean ± SEM of two groups unless otherwise stated, and significance is as follows: *$P < 0.05$; **$P < 0.01$; ***$P < 0.001$; and ****$P < 0.0001$. A $P$ value $> 0.05$ was considered nonsignificant (ns).

The number of replicates is indicated in each figure legend, except for figures depicting experiments performed once.

## Cell confluency and viability assays

For non-adherent AMO1, KMS27, and OPM2 cells, cells were plated at 5000 cells/well in ultra-low attachment (ULA) 96-well plates (Costar, #7007) and centrifuged at 600 × g for 5 min for spheroid formation. For adherent U2OS and Colo201 cells, 2000 cells/well were seeded in a clear flat-bottom 96-well plate (Costar, #3595). Treatments were added at the time of cell seeding, and cells were cultured in complete RPMI media to a final volume of 200 µL/well. Cultures were maintained at 37 °C throughout the duration of the experiment. Cell confluency was tracked using the Incucyte live-cell imaging system (Sartorius S3). Picture frames were captured every 4 h with a 4× objective, and cell confluency (%) from the total well area as a function of time was calculated using the Incucyte software (version 2023 A Rev2).

## Reporting summary

Further information on research design is available in the Nature Portfolio Reporting Summary linked to this article.

## Data availability

Coordinates of the IRE1:G6374:VCB complex have been deposited in the Protein Data Bank (PDB) with accession code 9N88. The corresponding electron microscopy density map has been deposited in the Electron Microscopy Data Bank (EMDB) under accession code EMD-49119. Other protein coordinates that have been used for model building or structural analysis are: 6W3B, 5T35, 3P23, 6W3K, 6URC, 8RX0, 8EWV, 7Q2J, 8BB5, 8BB4, 7JTO, 7JTP. Proteomics data have been deposited to ProteomeXChange Consortium via the MassIVE partner repository with the dataset identifier MSV000097222 [https://massive.ucsd.edu/ProteoSAFe/private-dataset.jsp?task=1aa403d6dc3647fb9157d230f9d7d2f6]. The authors declare that all other data supporting the findings of this study are available within the article and its Supplementary Data files, or from the corresponding authors on request. Source data are provided with this paper.

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

## Acknowledgements

The authors thank members of the Ashkenazi lab: Gladys Boenig, Georgette Castaneda, Caroline Gilchrist, Nick Endres, Robert Blake, Yiming Xu, Giovanni Luchetti, Claudio Ciferri, Jonas Tholen, Christopher Koo, Orlando Martinez, Anna Howes, Przemyslaw Dutka, Alister Burt, and Dimitry Tegunov for their contributions.

## Author contributions

J.D. and A.A. conceived and designed the study. J.D. performed the majority of experiments and analyzed data. K.C. performed the high-throughput PROTAC screening assay. M.-G.B. designed the IRE1 ligand portion of G6374. E.V. and J.R. conceived and generated G6374. M.H.B. designed and analyzed the SelectScreen Kinase Profiling assay. S.M. generated and characterized shIRE1 cell lines. J.D., D.L., and C.J.G. performed protein degradation and cell viability assays. S.F. guided and supported in vitro ubiquitination assays. K.C. guided and supported FP experiments. T.G. and S.R. performed and analyzed SPR experiments. T.K.C. and C.M.R. performed in vitro ubiquitination and global proteomics analyses by LC-MS/MS. M.J. guided and supported the preparation of cryoEM grids. M.J. and C.A. performed cryoEM grid screening and data collection, and guided image processing. J.D., P.H., and A.R. built and refined models. J.D., P.H., and A.A. wrote the manuscript with input from all authors. The overall project was supervised by J.R., P.H., and A.A.

## Competing interests

The following authors were employees of Genentech, Inc. at the time the work was performed: J.D., E.V., M.J., C.A., C.J.G., S.M., D.L., S.F., A.R., T.K.C., C.M.R., T.G., S.R., K.C., M.H.B., M.B., J.R., P.H., and A.A.
