## [Transparent Peer Review file · Nature Communications]

Chemically-induced degradation of the endoplasmic-reticulum stress sensor IRE1 by a VHL-recruiting chimera

Corresponding Author: Dr Avi Ashkenazi

Version 0:

Reviewer comments:

Reviewer #1

(Remarks to the Author)

The manuscript presented by Du et al. describes a PROTAC molecule targeting IRE1, a key protein that mediates the unfolded protein response in eukaryotic cells to maintain the homeostasis of their endoplasmic reticulum (ER). Although a previously described cereblon-based PROTAC molecule targeting IRE1 was reported recently—something that may reduce the novelty of the work presented here—the manuscript provides novel, timely, and very interesting biochemical and extremely interesting structural results. Specifically, this Reviewer would like to highlight the extensive work of the authors to elucidate the PROTAC-mediated IRE1 degradation pathway. However, in the opinion of this reviewer, the manuscript could be published in Nature Communications after some additional experiments to support certain results.

One of the most interesting findings of the manuscript is that VHL and IRE1 interact naturally in the absence of the PROTAC molecule. This observation raises an important question about the "real" contribution of the PROTACs to the degradation observed. In this context, and given the results, it is essential to add additional biophysical experiments to quantitatively assess binding and cooperativity. Affinity (K_d) should be measured for the initial IRE1 ligand employed, as well as for the PROTAC towards both VHL and IRE1. Moreover, the K_d of the VHL:IRE1 protein-protein interaction should be assessed. The discovery of G6374 is not well described. The IRE1-kinase ligands employed and the linkers used must be described and included in the manuscript. Additionally, the PROTAC structure-activity relationship (SAR) should be discussed for the benefit of the readers.

The specificity of G6374 against other kinases is well-assessed, and the conclusions drawn are highly informative—for example, the increased selectivity of the PROTAC compared to the initial ligand. Only MELK appears to be degraded by G6374 at similar concentrations. Could the authors discuss the possibility that the phenotypic effects observed in cells might also be due to MELK degradation? Additionally, proteomics experiments could provide insight into the selectivity of G6374 towards proteins outside the kinase family. Moreover, E3 ligase selectivity is not directly addressed in any of the immunoblotting experiments. It would be helpful to include an experiment assessing competitive treatment with the G6374 PROTAC molecule and the VHL ligand or to design a "negative" (i.e., cis-VHL) PROTAC molecule to probe selectivity toward the E3 ligase recruited by the PROTAC. In this context, the statement in line 144 is not fully corroborated by experimental data.

From a formatting perspective, the manuscript is well-written, organized, and easy to follow for readers of the journal. The figures and supplementary figures are clear and informative. However, in this reviewer's opinion, the chemical synthesis of G6374, at least the final compound, should be included in the Materials and Methods section. Additionally, the purity of the chemical compounds should be highlighted, particularly for G6374.

This Reviewer would also like to point out the following minor corrections/comments:

- The degradative capacity of the PROTAC on other potential (off-)targeted kinases (Fig. 1c and d) should ideally be tested at the same concentration and treatment time as the maximal degradation of IRE1, which has been demonstrated (at least 3 μM and 17 hours). At lower concentrations or shorter treatment times, the PROTAC might have reduced toxic effects, and thus the other kinases may not be degraded.
- Line 67: A reference should be included.
- Reference 28 is not about VHL ligands.
- Line 174: When the authors refer to four detectable sites, do they mean four surface-exposed lysines?
- In the Methods section (line 540), the column used to purify the protein is not specified.
- In the same section, the LC-MS data for phosphorylated and non-phosphorylated IRE1 are not shown; it may be useful to include these in the supplementary information.
- It is unclear where the proteins used in the in vitro ubiquitination experiment are sourced. Are they produced in-house, or

are they purchased from a vendor (if so, please specify which)? If they are produced in-house, details regarding the production protocol (expression and purification) are missing.

Reviewer #2

(Remarks to the Author)

Reviewer #3

(Remarks to the Author)

This paper describes the characterization of a VHL-recruiting IRE1 degrader in structural, biochemical and cell viability assays. Overall, it is a rigorous and novel body of work, which describes interesting 2:2 degradation complexes and enhanced degradation activity compared to existing probes. However, two key experiments are missing that would be important to address prior to publication.

1. Development of a negative control degrader, using a diastereomer or the VHL ligand that can no longer bind VHL, and no longer degrade IRE1 is needed in phenotypic cell proliferation assays to demonstrate what proportion of the anti-proliferative effects are degradation dependent, as the lead the compound also retains on and off target kinase inhibition pharmacology. Inclusion of the parent inhibitor in these assays is also desirable.
2. Global proteomics analysis is required to demonstrate selectivity of the lead degrader, as low affinity binding can translate into potent degradation in the case of co-operative complexes.

Reviewer #4

(Remarks to the Author)

The authors elegantly and clearly present a novel PROTAC degrader molecule called G6374, which recruits the CRL2VHL E3 ligase for the proteasomal degradation of the inositol-requiring enzyme 1 (IRE1).

The authors firstly compare IRE1 degradation in AMO1 MM cells and show that G6374 is more effective than the previously published IRE1 degrader CPD-2828 after a 17-hour treatment, although it is unclear how many replicates were performed, or if there were any other timepoints. Selectivity against a library of 220 kinases was assessed, and G6374 appears to recruit several kinases. However, degradation was assessed for the kinases which were inhibited at a level of 90%, and no degradation was observed. Using an E1 inhibitor and a neddylation inhibitor, the authors show the degradation pathway is cullin RING and UPS-dependent.

K48-linkages are highlighted by western blot, confirming that polyubiquitin chains signalling for proteasomal degradation can be constructed on the target protein IRE1. Ubiquitination was also performed in vitro, using UBE2R1, which is known to work with CUL2 and also can mediate K48-chain formation. While UBE2R1 is known to be proficient at extending polyubiquitin chains, in this system it appears that UBE2R1 is also capable of installing the first ubiquitin on IRE1. The UBE2D family of proteins does not appear to be used in this study. The authors also show that ubiquitination can occur independent of IRE1 phosphorylation status in vitro.

To investigate whether G6374 can form a ternary complex in cells, the authors perform a pull-down assay and show association with endogenous VHL by western blot. A cellular NanoBRET assay is also performed, and the ternary complex also forms in vitro using recombinant proteins. These orthogonal assays demonstrate unambiguously that IRE1 ubiquitination and degradation is driven by G6374 ternary complex formation with IRE1 and VHL. An excellent addition to the study is the cryo-EM structure, which reveals the stoichiometry of the interaction. A 2:2 IRE1:VHL ternary complex is revealed and IRE1 forms 'back to back' dimers, as seen previously in the literature. The authors solve a 2.6 Å structure of IRE1-G6374-VHL-EloC. EloB was also present in the sample, but owing to its lower resolution, the authors elect not to include this in the atomic model, although a few residues are visible. Pleasingly, the compound G6374 can be seen unambiguously and the IRE1-VHL interface is well-resolved. IRE1 atomic model aligns well with previously published back-to-back structures. Based on this cryo-EM structure, point mutants of NanoLuc-IRE1 were generated and showed that the key induced protein-protein and protein-compound interactions highlighted from the cryo-EM structure can be validated in cells. The expression levels of these constructs are also validated by IB, showing that the mutants maintain their stability and the lack of ternary complex binding is likely due to the point mutations alone.

The authors show that using G6374, IRE1 can be fully degraded after a 17-hour treatment, which is an improvement on the previously published CPD-2828. The authors also show that IRE1 degradation mediated by G6374 treatment blocks cancer cell proliferation.

Overall, the paper is significant and reports an exciting new VHL-recruiting IRE1 degrader molecule, and it reveals insights on how this compound mediated ternary complex formation and ubiquitination. The manuscript is clearly written, flows logically, and advances our understanding of how IRE1 can be targeted for degradation with small molecule degraders. This is an interesting and impactful study which, in our opinions, should eventually be published subject to some revisions

outlined below.

1) Figure 1b: It is unclear how many replicates were performed. If providing quantification and comparing efficacy against a previously published degrader, more than one replicate should be included. Given that one of the main claims of the manuscript is that G6374 mediates 'complete degradation' of IRE1, it would be worthwhile comparing G6374 with the previously published degrader at a shorter treatment time or with a timecourse experiment, to adequately benchmark G6374.

2) The same comment as 2) applies to many of the experiments performed in the manuscript. It is unclear whether biological replicates have been performed. There is little indication of whether the results obtained are statistically relevant when comparing results between species and groups.

3) Page 5 line 130: It does seem reasonable to state that compound binding to the substrate, and therefore kinase inhibition, may not correlate exactly with kinase degradation. Would this therefore mean that any inhibition observed in Figure 1c could potentially lead to degradation? If this is the case, then the experiment assesses the selectivity of G6374 in recruiting IRE1 over other kinases, but does not establish G6374 as a selective degrader, as any kinase showing inhibition could be degraded. If the authors wish to make claims on degrader selectivity, then it is likely that a proteomics experiment would be required to appreciate any off-target effects in terms of degradation.

4) Page 6 lines 168-182: Could the authors specify in the main text and also in the legend of Figure 1 which E2 is used for the Me-Ub reaction, and provide some verbiage as to why this E2 was selected? From reading the methods it appears this is UBE2R1. Are the ubiquitination sites different when an E2 from the UBE2D family is used, which are also known to work with CRL2VHL, ubiquitinate PROTAC-recruited substrates and can also build K48 and K11 chains? See Supplementary Information Fig S6 and S7: <https://www.science.org/doi/10.1126/sciadv.ado6492#core-R73>

5) The authors highlight two lysines which show evidence of ubiquitination in vitro by UBE2R1, and also perform atomic model alignments to evaluate the feasibility of ubiquitination. It appears that K704 is within reach, but K717 is on the opposite face of IRE1. When modelling the IRE2 dimer with CRL2VHL monomer, it appears that K717 is now more readily accessible. Could the other lysines on IRE1 which did not show evidence of being ubiquitinated also be modelled, to see whether these are located on the same 'face' of IRE1 accessible to the E2 catalytic site? Could the authors also model the IRE1-VHL dimer in the presence of two copies of CRL2, i.e. the IRE12-CRL2VHL2 dimer? To validate the relevance of these two lysine residues, the authors could generate the K704R and K717R single and double mutants and assess their reactivity.

Minor comments:

6) Page 1 line 27: VHL is the substrate receptor of the CRL2VHL complex, rather than an adapter protein.

7) Page 4 line 107: Is there a reason why the other, less efficient, compounds in the panel are not shown? Could this be shown in the SI? It would be interesting to see the SAR work leading to the discovery of G6374.

8) Page 3 line 86: I question the relevance of references 26 and 27 here. The VH032 ligand for VHL was developed by Galdeano et al, J Med Chem, 2014 and then used as a PROTAC in Zengerle et al ACS Chem Biol 2015.

9) Figure 1d: The compound name G6374 has changed to just '6374', which could confuse the reader.

10) Figure 1c/d: does the timepoint used for the kinase inhibition experiment match the treatment time for the degradation assay with the kinase panel? It would be good to specify the treatment times in the figure legends.

11) Figure 1d: An issue with western blotting seems to have occurred for PKD1 identification in KMS27, and the resulting blot is not of high enough quality to clearly show a lack of PKD1 degradation.

12) Page 6 line 160: the E2 ubiquitin-conjugating enzyme could also be mentioned at this point. Is there a reason why UBE2R1 (Cdc34) was used rather than a protein from the UBE2D family for in vitro experiments?

13) Accession codes for the proteomics data appear to be missing. Will the results be deposited?

14) Supplementary Figure 1e, f, g: It is unclear what each band 01-04 corresponds to. Could a theoretical MW or IRE-Ub1, IRE-Ub2, IRE-Ub3 etc. be assigned for each band on the Coomassie stain figures? This should at least be possible for the Me-Ub where the bands are more distinguished. For clarity for the readers, could the authors also annotate the strongly stained bands below the 37 kDa marker and between the 150-100 kDa markers which presumably are UBE2R1 and UBE1? It is unclear if Supplementary Figure 1 f counts is a combination of all samples from bands 01-04, or just corresponds to a single excised band.

15) Supplementary Figure 1i: Could the authors also highlight any lysines which did not show evidence of being modified by UBE2R1?

16) Page 8: Could the authors mention in the main text that the cryo-EM complex was cross-linked with BS3, and why this

was necessary? This is nicely justified in the materials and methods section (page 20 line 620), but this could be brought to the main text. Was the sample used for crosslinking and cryo-EM grid preparation the fully-formed SEC product?

17) Out of curiosity, why is it unexpected that VHL engages the B2B dimer (page 8 line 228)?

18) According to Supplementary Figure 3e the cryo-EM volume appears to have a good range of distributions. Would it be possible to include the directional FSCs and the sphericity values from the 3D FSC server? <https://3dfsc.salk.edu> Many 2D classes are shown, would it be possible to show fewer but to enlarge these so they can be better appreciate? For additional cryo-EM data validation, the authors could consider including the EMRinger score.

19) Rather than removing EloB altogether, residues 68-73 are well-resolved with density for side-chains and could be included in the deposited atomic model, to help show unambiguously that EloB is present in the sample. However, certain parts of VHL and EloC are not visible in the supplied sharpened map and therefore could be truncated from the atomic model.

20) In Figure 4b-c-e-f, the figure legend states that the NanoBRET data for the wild-type IRE1 is the same as is shown in Fig. 2. Could the authors please clarify whether the experiment with the wild-type IRE1 was performed at the same time/in parallel to the mutants, to ensure that the lack of signal from the IRE1 mutants is due to the mutations themselves rather than any experimental issue, and to ensure a fair comparison between the wild-type and mutants?

21) Page 9 line 280. In order to make strong claims such as 'dramatically reduced', it would be good to include whether the difference between the DC50 and Dmax values are statistically significant? If there does exist a statistically significant difference, then a more scholarly term such as 'significantly reduced' could be included.

22) Page 10. The authors may wish to acknowledge that the E2 in the model presented in Figure 5 is different to the one used in the in vitro ubiquitination assays (UBE2R1/Cdc34 versus UBE2R2/Cdc34b). A structure of VHL-EloB-EloC-(NEDD8)-CUL2-RBX1-UBE2R1-Ub has also been published (PDB: 8RX0).

23) Several typos/inconsistencies in the methods which should be corrected prior to publication.

24) As the authors will be aware, another IRE1 degrader was recently published. This should be appropriately referenced in the manuscript: <https://pubs.rsc.org/en/content/articlelanding/2025/md/d5md00028a>.

25) As the authors will be aware, cryo-EM structures of whole CRL2-VHL complexes with E2-Ub and neo-substrates have been recently disclosed and their relevant to ubiquitination specificity have been investigated by us (<https://doi.org/10.1126/sciadv.ado6492>) and the Schulman/Kleiger Labs (<https://doi.org/10.1016/j.molcel.2024.01.022>), which should be duly discussed and cited.

We agree to waive our anonymity in a spirit to enhance the transparency of the peer review process, Charlotte Crowe & Alessio Ciulli

Version 1:

Reviewer comments:

Reviewer #1

(Remarks to the Author)

The authors have satisfactorily addressed all the concerns raised in the previous reviews and have conducted the additional experiments as requested by the reviewers, significantly strengthening the manuscript and increasing the robustness of the results and the conclusions.

In the opinion of this Reviewer, the manuscript is ready for publication in Nature Communications after two minor additions.

Which high-throughput degradation assay was used to identify G6374 as the most potent PROTAC? The results (data) from this experiment should be included in the Supporting Information, at least in the form of a graphic showing the degradation level achieved by each degraded synthesised in the library. Moreover, the methodological aspects of this experiment should be included in the Materials and Methods Section.

In parallel, the authors used in this revision a Fluorescence Polarization experiment to show cooperativity. It would be useful for the readers to mention in the manuscript if this biophysical approach has been used for this purpose, and add, if possible, references.

Reviewer #2

(Remarks to the Author)

I co-reviewed this manuscript with one of the reviewers who provided the listed reports. This is part of the Nature

Communications initiative to facilitate training in peer review and to provide appropriate recognition for Early Career Researchers who co-review manuscripts.

Reviewer #3

(Remarks to the Author)

The authors have rigorously addressed all my concerns and I am now happy to recommend publication. Congratulations on the nice body of work!

Reviewer #4

(Remarks to the Author)

The authors have taken on board all comments, performed all of the suggested additional experiments and analyses, and have rigorously addressed all point made in their response and also in the main manuscript. The manuscript and the claims made therein have been significantly strengthened.

Notably, the degradation data has been strengthened by including timecourse treatments and biological replicates with statistical analyses. Further mode of action validation has been performed with the cis-VHL analogue PROTAC and siRNA of VHL, showing a VHL-dependent mechanism of degradation [this is in addition to the previous mode of action validation showing CRL and UPS-dependence]. A new proteomics experiment shows kinase selectivity. Another excellent addition is the biophysical characterisation (by SPR and FP) of the binary and ternary complexes, which shows that the PROTAC ternary complex is indeed cooperative.

The cryo-EM structure is of excellent resolution and coupled with the ubiquitination assays provides deep insight into the mechanism of IRE1 ubiquitination. The sample preparation for the cryo-EM has been clarified and brought up-front, which should help and guide the TPD community with solving more degrader ternary complexes. Additional studies with UBE2D1 and UBE2D2 have been performed and the results have been compared and contrasted with UBE2R1. Site-directed mutagenesis of the ubiquitinated lysines has also been performed, and discussion on this topic has been provided. These additional data, in conjunction with the IRE1 monomer and B2B dimer modelling with the full CRL2 scaffold fully rationalise why certain residues on IRE1 are targeted for ubiquitination.

Overall, the paper is significant and reports an exciting new VHL-recruiting IRE1 degrader molecule. Furthermore, it provides structural and mechanistic insights into the ubiquitination of IRE1 by CRL2VHL, which should be of great interest to the TPD field and contributes to advancing our knowledge about ubiquitination in TPD. We recommend that this manuscript should be published.

In a spirit to enhance transparency in the peer review process, we agree to waive our anonymity. Charlotte Crowe & Alessio Ciulli

Open Access This Peer Review File is licensed under a Creative Commons Attribution 4.0 International License, which permits use, sharing, adaptation, distribution and reproduction in any medium or format, as long as you give appropriate credit to the original author(s) and the source, provide a link to the Creative Commons license, and indicate if changes were

made.

Reviewer #1 (Remarks to the Author)

The manuscript presented by Du et al. describes a PROTAC molecule targeting IRE1, a key protein that mediates the unfolded protein response in eukaryotic cells to maintain the homeostasis of their endoplasmic reticulum (ER). Although a previously described cereblon-based PROTAC molecule targeting IRE1 was reported recently—something that may reduce the novelty of the work presented here—the manuscript provides novel, timely, and very interesting biochemical and extremely interesting structural results. Specifically, this Reviewer would like to highlight the extensive work of the authors to elucidate the PROTAC-mediated IRE1 degradation pathway. However, in the opinion of this reviewer, the manuscript could be published in Nature Communications after some additional experiments to support certain results.

We thank the reviewer for this positive assessment of our work and for the constructive suggestions.

One of the most interesting findings of the manuscript is that VHL and IRE1 interact naturally in the absence of the PROTAC molecule. This observation raises an important question about the "real" contribution of the PROTACs to the degradation observed. In this context, and given the results, it is essential to add additional biophysical experiments to quantitatively assess binding and cooperativity. Affinity (K_d) should be measured for the initial IRE1 ligand employed, as well as for the PROTAC towards both VHL and IRE1. Moreover, the K_d of the VHL:IRE1 protein-protein interaction should be assessed.

As suggested, we have performed additional biophysical assays using surface plasmon resonance (SPR) to measure the K_d values between IRE1 and VHL, IRE1 and the PROTAC G6374, VHL and the PROTAC G6374, as well as IRE1 and the parent IRE1 ligand G3201, and VHL and the parent VHL ligand VH032. A Table summarizing these values is now included (**Supplementary Table 1**).

	Complex measured	$k_a / M^{-1}s^{-1}$	k_d / s^{-1}	K_d / nM	IC ₅₀ (nM)	Cooperativity alpha
SPR	IRE1: VCB	N/A	N/A	>10000 (N=2)	N/A	N/A
	IRE1: G6374	4.95E+06 ± 2.85E+06 (N=2)	2.50E-03 ± 1.20E-03 (N=2)	0.56 ± 0.09 (N=2)	N/A	N/A
	VCB: G6374	5.14E+05 ± 2.24E+04 (N=3)	4.77E-02 ± 5.44E-03 (N=3)	90.23 ± 4.98 (N=3)	N/A	N/A
	IRE1: G3201	5.30E+06 (N=1)	1.42E-02 (N=1)	2.70 (N=1)	N/A	N/A
	VCB: VH032	5.50E+05 (N=1)	1.30E-01 (N=1)	232.00 (N=1)	N/A	N/A
FP	VCB: G6374	N/A	N/A	N/A	2476 ± 112 (N=3)	5 ± 0.15 (N=3)
	VCB: (G6374+IRE1)	N/A	N/A	N/A	492 ± 21 (N=3)	

We also investigated whether G6374 induces ternary complex formation in cooperative fashion. To this end, we used fluorescence polarization (FP), which is performed in solution, rather than SPR, which is prone to artifacts due to potential IRE1 oligomerization in a semi-immobilized setting. As shown below and included in **Fig. 2f**, FP indicated that G6374 induces ternary complex formation in cooperative manner.

The discovery of G6374 is not well described. The IRE1-kinase ligands employed and the linkers used must be described and included in the manuscript. Additionally, the PROTAC structure-activity relationship (SAR) should be discussed for the benefit of the readers.

As requested, we have included additional information in the beginning of the Results section regarding the discovery and evolution of G6374 as shown schematically here below and now included in **Supplementary Fig. 1a**.

The specificity of G6374 against other kinases is well-assessed, and the conclusions drawn are highly informative—for example, the increased selectivity of the PROTAC compared to the initial ligand. Only MELK appears to be degraded by G6374 at similar concentrations. Could the authors discuss the possibility that the phenotypic effects observed in cells might also be due to MELK degradation? Additionally, proteomics experiments could provide insight into the selectivity of G6374 towards proteins outside the kinase family.

Following a general request from reviewer #4, we performed several additional biological replicates of the analysis in **Fig. 1d**. This revealed that the initially suspected depletion of MELK in AMO1 cells was an outlier, while three additional experiments showed that MELK is not depleted in response to G6374.

Nevertheless, to further ascertain that MELK is not involved in the growth inhibition by G6374, we tested the established MELK inhibitor OTSSP167, which blocks MELK activity with an IC_{50} of 0.4 nM. As shown below, even at 3X the IC_{50} OTSSP167 did not affect proliferation of AMO1 cells whereas IRE1 knockdown completely blocked growth. Accordingly, we no longer refer to MELK in the text as a functionally relevant potential off-target.

As suggested, we generated an epimer control compound, G2642, which is illustrated below. We functionally verified that G2642 does not deplete IRE1, as also shown below and now included in **Supplementary Fig. 1 e,f**. We then performed a proteomics analysis comparing G6374 and G2642, as shown below and now depicted in **Fig. 1e**, which confirmed that IRE1 is the most substantially and significantly depleted protein target of G6374.

Moreover, E3 ligase selectivity is not directly addressed in any of the immunoblotting experiments. It would be helpful to include an experiment assessing competitive treatment with the G6374 PROTAC molecule and the VHL ligand or to design a “negative” (i.e., cis-VHL) PROTAC molecule to probe selectivity toward the E3 ligase recruited by the PROTAC. In this context, the statement in line 144 is not fully corroborated by experimental data.

As noted above, we confirmed that the epimer G2642 does not degrade IRE1 (**Supplementary Fig. 1f**). To further establish the dependency on VHL, we performed siRNA-based VHL knockdown experiments. We used HCT116 cells because they are more efficiently transfectable than AMO1 cells. As shown below, although VHL knockdown by itself slightly lowered baseline IRE1 levels, it essentially prevented G6374-induced IRE1 depletion. This data is now depicted in **Supplementary Fig. 1h**.

From a formatting perspective, the manuscript is well-written, organized, and easy to follow for readers of the journal. The figures and supplementary figures are clear and informative. However, in this reviewer's opinion, the chemical synthesis of G6374, at least the final compound, should be included in the Materials and Methods section.

We appreciate this kind feedback. A description of the chemical synthesis of G6374 was included in the **Supplementary Information Section**.

Additionally, the purity of the chemical compounds should be highlighted, particularly for G6374.

We have added the NMR chromatograms for the compounds to the chemical synthesis section.

This Reviewer would also like to point out the following minor corrections/comments:

- The degradative capacity of the PROTAC on other potential (off-)targeted kinases (Fig. 1c and d) should ideally be tested at the same concentration and treatment time as the maximal degradation of IRE1, which has been demonstrated. At lower concentrations or shorter treatment times, the PROTAC might have reduced toxic effects, and thus the other kinases may not be degraded.

As shown below and now included in **Supplementary Fig. 1b**, in a time-course experiment, G6374-induced IRE1 depletion reaches near maximum levels by 4 hr at 1 μM. On the other hand, longer incubations may lead to indirect, secondary effects of the primary depletion of IRE1 itself. Therefore, we believe that the conditions we chose represent a reasonable balance between direct IRE1 degradation and potential indirect secondary effects on other proteins.

- Line 67: A reference should be included.

Thank you for pointing this out. We have added the reference.

- Reference 28 is not about VHL ligands.

Thank you for pointing this out. We have corrected the references.

- Line 174: When the authors refer to four detectable sites, do they mean four surface-exposed lysines?

We are simply using cautious language because the LC-MS/MS technique used may fail to detect modifications on peptides that are not sufficiently abundant.

- In the Methods section (line 540), the column used to purify the protein is not specified.

Thank you for pointing this out. We added the missing information.

- In the same section, the LC-MS data for phosphorylated and non-phosphorylated IRE1 are not shown; it may be useful to include these in the supplementary information.

Thank you for this suggestion. We have included the LC-MS data for IRE1.KR 0P and 3P to **Supplementary Fig. 1i**.

- It is unclear where the proteins used in the *in vitro* ubiquitination experiment are sourced. Are they produced in-house, or are they purchased from a vendor (if so, please specify which)? If they are produced in-house, details regarding the production protocol (expression and purification) are missing.

We have added the following information regarding proteins used in the *in vitro* ubiquitination assay to the **Materials and Methods** section.

Reviewer #2 (Remarks to the Author)

Thank you for reviewing the manuscript and for providing insightful and constructive comments. We hope this was a productive use of your time! Please see our replies to Reviewer #4 below.

Reviewer #3 (Remarks to the Author)

This paper describes the characterization of a VHL-recruiting IRE1 degrader in structural, biochemical and cell viability assays. Overall, it is a rigorous and novel body of work, which describes interesting 2:2 degradation complexes and enhanced degradation activity compared to existing probes. However, two key experiments are missing that would be important to address prior to publication.

We thank the reviewer for the positive assessment of our work and for the constructive suggestions on how to improve it.

1. Development of a negative control degrader, using a diastereomer or the VHL ligand that can no longer bind VHL, and no longer degrade IRE1 is needed in phenotypic cell proliferation assays to demonstrate what proportion of the anti-proliferative effects are effects are degradation dependent, as the lead the compound also retains on and off target kinase inhibition pharmacology. Inclusion of the parent inhibitor in these assays is also desirable.

Thank you for this important suggestion. As noted in our reply to reviewer #1 above, we have generated an epimer (diastereomer) control, G2642, and confirmed that it does not deplete IRE (Supplementary Fig. 1 e,f).

We further confirmed that the epimer has no anti-proliferative activity against IRE1-dependent or -independent cell lines, as shown below and now included in Supplementary Fig. 6 f-j.

2. Global proteomics analysis is required to demonstrate selectivity of the lead degrader, as low affinity binding can translate into potent degradation in the case of co-operative complexes.

As noted above in our reply to Reviewer #1, we have performed a proteomics analysis comparing G6374 with G2642. This analysis, now shown in **Fig. 1e**, confirms that IRE1 is the most substantially and significantly depleted protein target of G6374.

Reviewer #4 (Remarks to the Author):

The authors elegantly and clearly present a novel PROTAC degrader molecule called G6374, which recruits the CRL2VHL E3 ligase for the proteasomal degradation of the inositol-requiring enzyme 1 (IRE1).

The authors firstly compare IRE1 degradation in AMO1 MM cells and show that G6374 is more effective than the previously published IRE1 degrader CPD-2828 after a 17-hour treatment, although it is unclear how many replicates were performed, or if there were any other timepoints. Selectivity against a library of 220 kinases was assessed, and G6374 appears to recruit several kinases. However, degradation was assessed for the kinases which were inhibited at a level of 90%, and no degradation was observed. Using an E1 inhibitor and a neddylation inhibitor, the authors show the degradation pathway is cullin RING and UPS-dependent.

K48-linkages are highlighted by western blot, confirming that polyubiquitin chains signalling for proteasomal degradation can be constructed on the target protein IRE1. Ubiquitination was also performed in vitro, using UBE2R1, which is known to work with CUL2 and also can mediate K48-chain formation. While UBE2R1 is known to be proficient at extending polyubiquitin chains, in this system it appears that UBE2R1 is also capable of installing the first ubiquitin on IRE1. The UBE2D family of proteins does not appear to be used in this study. The authors also show that ubiquitination can occur independent of IRE1 phosphorylation status in vitro.

To investigate whether G6374 can form a ternary complex in cells, the authors perform a pull-down assay and show association with endogenous VHL by western blot. A cellular NanoBRET assay is also performed, and the ternary complex also forms in vitro using recombinant proteins. These orthogonal assays demonstrate unambiguously that IRE1 ubiquitination and degradation is driven by G6374 ternary complex formation with IRE1 and VHL. An excellent addition to the study is the cryo-EM structure, which reveals the stoichiometry of the interaction. A 2:2 IRE1:VHL ternary complex is revealed and IRE1 forms 'back to back' dimers, as seen previously in the literature. The authors solve a 2.6 Å structure of IRE1- G6374-VHL-EloC. EloB was also present in the sample, but owing to its lower resolution, the authors elect not to include this in the atomic model, although a few residues are visible. Pleasingly, the compound G6374 can be seen unambiguously and the IRE1-VHL interface is well-resolved. IRE1 atomic model aligns well with previously published back-to-back structures. Based on this cryo-EM structure, point mutants of NanoLuc-IRE1 were generated and showed that the key induced protein-protein and protein-compound interactions highlighted from the cryo-EM structure can be validated in cells. The expression levels of these constructs are also

validated by IB, showing that the mutants maintain their stability and the lack of ternary complex binding is likely due to the point mutations alone.

The authors show that using G6374, IRE1 can be fully degraded after a 17-hour treatment, which is an improvement on the previously published CPD-2828. The authors also show that IRE1 degradation mediated by G6374 treatment blocks cancer cell proliferation.

Overall, the paper is significant and reports an exciting new VHL-recruiting IRE1 degrader molecule, and it reveals insights on how this compound mediated ternary complex formation and ubiquitination. The manuscript is clearly written, flows logically, and advances our understanding of how IRE1 can be targeted for degradation with small molecule degraders. This is an interesting and impactful study which, in our opinions, should eventually be published subject to some revisions outlined below.

We thank the reviewers for their thorough review and positive assessment of our work and for their insightful and constructive suggestions on how to improve the manuscript.

1) Figure 1b: It is unclear how many replicates were performed. If providing quantification and comparing efficacy against a previously published degrader, more than one replicate should be included. Given that one of the main claims of the manuscript is that G6374 mediates 'complete degradation' of IRE1, it would be worthwhile comparing G6374 with the previously published degrader at a shorter treatment time or with a time course experiment, to adequately benchmark G6374.

As requested, we have performed additional biological replicates for the titration and the time-course experiments with G6374, as shown below. New graphs including error bars are now shown in **Fig. 1b** and **Supplementary Fig. 1 b,c**.

original figure:

n=3 biological replicates, mean ± SEM

2) The same comment as 2) applies to many of the experiments performed in the manuscript. It is unclear whether biological replicates have been performed. There is little indication of whether the results obtained are statistically relevant when comparing results between species and groups.

We have performed biological replicates for **Fig. 4 g,h**, **Fig. 5b**, and **Supplementary Fig. 1h**. We have calculated p-values where appropriate to confirm whether the differences are statistically significant.

3) Page 5 line 130: It does seem reasonable to state that compound binding to the substrate, and therefore kinase inhibition, may not correlate exactly with kinase degradation. Would this therefore mean that any inhibition observed in Figure 1c could potentially lead to degradation? If this is the case, then the experiment assesses the selectivity of G6374 in recruiting IRE1 over other kinases, but does not establish G6374 as a selective degrader, as any kinase showing inhibition could be degraded. If the authors wish to make claims on degrader selectivity, then it is likely that a proteomics experiment would be required to appreciate any off-target effects in terms of degradation.

As noted above in our reply to the other reviewers, we have performed a proteomics analysis comparing G6374 to an epimer control, G2642, which does not degrade IRE1. This analysis, now shown in **Fig. 1e**, confirms that IRE1 is the most substantially and significantly depleted target of G6374. Furthermore, as noted above in the response to Reviewer #3, we have verified that the epimer control has no anti-proliferative activity against IRE1-dependent or -independent cell lines.

4) Page 6 lines 168-182: Could the authors specify in the main text and also in the legend of Figure 1 which E2 is used for the Me-Ub reaction, and provide some verbiage as to why this E2 was selected? From reading the methods it appears this is UBE2R1. Are the ubiquitination sites different when an E2 from the UBE2D family is used, which are also known to work with CRL2VHL, ubiquitinate PROTAC-recruited substrates and can also build K48 and K11 chains? See Supplementary Information Fig S6 and S7: <https://www.science.org/doi/10.1126/sciadv.ado6492#core-R73>

Thank you for this request. As now specified in the text, we used UBE2R1 (CDC34), based on its previous use in validation studies of a VHL-based PROTAC (<https://www.sciencedirect.com/science/article/pii/S2451945623000302?via%3Dihub>).

We also confirmed via the Depmap portal (<https://depmap.org/portal>) that UBE2R1 and UBE2D1 and UBE2D2 have similar expression levels in model cancer cell lines that depend on IRE1's nonenzymatic activity, as shown below.

As also suggested, we performed the *in vitro* ubiquitination assay with additional E2 enzymes. As shown below and now included in **Supplementary Fig. 1 r,s**, the E2 enzymes UBE2D1 and UBE2D2 have a broader linkage propensity than UBE2R1, which is heavily influenced by the E3 ligase partner (Ye et al., Nat Rev Mol Cell Biol (2009); Gazdoui et al, Molecular and Cellular Biology (2007); Choi et al, JBC (2015)). Whereas UBE2R1 primarily generated K48-linked polyubiquitination of IRE1.KR, the UBE2D1 and UBE2D2 proteins predominantly generated K11-linked polyubiquitination.

As regards to the number of ubiquitination sites, UBE2D2 was more distinct from UBE2R1 in that it primarily ubiquitinated one lysine on IRE1.KR. By comparison, UBE2D1 showed a more intermediate activity in that it ubiquitinated a second site less efficiently than UBE2R1, yet more efficiently than UBE2D2 (**Supplementary Fig. 1s**). LC-MS/MS identified K704 as the predominant site on IRE1 that is ubiquitinated with UBE2D1 as well as UBE2D2 (**Supplementary Fig. 1 t,u**). The slight difference in ubiquitination site selection may be explained by the tendency of distinct E2s to prime their substrate differently, which can impact the specificity and efficiency of ubiquitination (Li et al, MolCell (2024); Crowe et al., Sci Adv (2024); also see in the **Discussion** section).

5) The authors highlight two lysines which show evidence of ubiquitination *in vitro* by UBE2R1, and also perform atomic model alignments to evaluate the feasibility of ubiquitination. It appears that K704 is within reach, but K717 is on the opposite face of IRE1. When modelling the IRE2 dimer with CRL2VHL monomer, it appears that K717 is now more readily accessible. Could the other lysines on IRE1 which did not show evidence of being ubiquitinated also be modelled, to see whether these are located on the same 'face' of IRE1 accessible to the E2 catalytic site? Could the authors also

model the IRE1-VHL dimer in the presence of two copies of CRL2, i.e. the IRE12-CRL2VHL2 dimer? To validate the relevance of these two lysine residues, the authors could generate the K704R and K717R single and double mutants and assess their reactivity.

By highlighting all the lysine residues on IRE1.KR (purple except K704 and K717, which are colored in red), K656, K799, K871 (on IRE1.KR that is directly associated with CRL2^{VHL}) and K716, K748, K811, K819, K824 (on the IRE1.KR protomer that does not directly associate with CRL2^{VHL}) appear to be accessible to Ub transfer. Our mass-spectrometry experiment showed that only K704 and K717 are mapped as ubiquitination sites with confidence. A plausible explanation for this apparent site restriction is that the addition of polyubiquitin chains at preferred lysine residues hinders modification at additional positions on IRE1. Consistent with this notion, mutation of K704 and K717 to arginine did not abolish ubiquitination on IRE1; rather, it altered the ubiquitination pattern as detected with methyl-ubiquitin conjugation (**Supplementary Fig. 7 a,b**). This is now noted in the **Discussion**.

Modeling the IRE1-VHL dimer in the presence of two copies of CRL2^{VHL} shows that the K704 residues in each IRE1 protomer can be ubiquitinated by the CRL2 complex that is directly associated through VHL, and the K717 residues can be ubiquitinated by the CRL2 complex that is associated with its protomer in the IRE1 B2B dimer.

We generated the suggested IRE1 mutant proteins and performed *in vitro* ubiquitination assays with IRE1.KR WT, K704R, K717R and K704R/K717R. We found that the mutants can still be polyubiquitinated. This is not surprising, as CRL2 is known to be able to capture suboptimal lysines (Crowe *et al.*, *Sci Adv.* (2024)). Indeed, in our own lab's prior experience (Jin *et al.*, *Cell* (2009)), in a study of CRL-mediated caspase-8 ubiquitination we observed that mutation of the most preferred ubiquitination sites resulted in new sites being unmasked – which necessitated mutating all 6 lysines to abolish ubiquitination altogether. Furthermore, as shown below, and now included **Supplementary Fig.1 p,q**, *in vitro* ubiquitination results with Methyl-Ub showed that, distinct from ubiquitinated IRE1.KR WT where predominantly 2 different lysines are ubiquitinated, the ubiquitinated IRE1 mutants had higher percentages with 3 lysines being monoubiquitinated, suggesting that other lysines become preferred ubiquitination sites once steric hindrance of K704 and K717 is removed.

Minor comments:

6) Page 1 line 27: VHL is the substrate receptor of the CRL2VHL complex, rather than an adapter protein.

Thank you for pointing this out. We have corrected the text.

7) Page 4 line 107: Is there a reason why the other, less efficient, compounds in the panel are not shown? Could this be shown in the SI? It would be interesting to see the SAR work leading to the discovery of G6374.

We have included a paragraph that discusses the evolution of G6374 in the **Results** section.

8) Page 3 line 86: I question the relevance of references 26 and 27 here. The VH032 ligand for VHL was developed by Galdeano et al, J Med Chem, 2014 and then used as a PROTAC in Zengerle et al ACS Chem Biol 2015.

Thank you for pointing this out. We have corrected the references.

9) Figure 1d: The compound name G6374 has changed to just '6374', which could confuse the reader.

Thank you for catching this error. It has been corrected.

10) Figure 1c/d: does the timepoint used for the kinase inhibition experiment match the treatment time for the degradation assay with the kinase panel? It would be good to specify the treatment times in the figure legends.

The SelectScreen Kinase Profiling was performed at 1 uM for 1 hr, which was originally stated in the methods section, and now also included the figure legend.

11) Figure 1d: An issue with western blotting seems to have occurred for PKD1 identification in KMS27, and the resulting blot is not of high enough quality to clearly show a lack of PKD1 degradation.

We performed the degradation assay in KMS27 with N=3 and confirmed that there is no degradation of PKD1.

12) Page 6 line 160: the E2 ubiquitin-conjugating enzyme could also be mentioned at this point. Is there a reason why UBE2R1 (Cdc34) was used rather than a protein from the UBE2D family for in vitro experiments?

As noted in our reply to major comment 4) above, we used UBE2R1 (CDC34) because it has been used in other VHL-based PROTAC validation (<https://www.sciencedirect.com/science/article/pii/S2451945623000302?via%3Dihub>).

13) Accession codes for the proteomics data appear to be missing. Will the results be deposited?

Yes, the proteomics dataset has been deposited in MassIVE, with code MSV000097222. Currently it requires access with password “IRE1418” but will become publicly available upon publication. We have included this information in the “Data availability” section in the revised manuscript.

14) Supplementary Figure 1e, f, g: It is unclear what each band 01-04 corresponds to. Could a theoretical MW or IRE-Ub1, IRE-Ub2, IRE-Ub3 etc. be assigned for each band on the Coomassie stain figures? This should at least be possible for the Me-Ub where the bands are more distinguished. For clarity for the readers, could the authors also annotate the strongly stained bands below the 37 kDa marker and between the 150-100 kDa markers which presumably are UBE2R1 and UBE1? It is unclear if Supplementary Figure 1 f counts is a combination of all samples from bands 01-04, or just corresponds to a single excised band.

The result corresponds to a combination of all bands. We have removed the lines in the boxes to avoid confusion.

The molecular weights for recombinant proteins used in the *in vitro* ubiquitination assay and the theoretical molecular weights for ubiquitinated IRE1 are listed below:

Proteins	M.W. (kDa)
UBE 1	118
CUL2	87
IRE1.KR-4Ub	84.3
IRE1.KR-3Ub	75.6
IRE1.KR-2Ub	66.9
IRE1.KR-1Ub	58.2
UBA3	52
IRE1.KR-0Ub	49.5
UBE2R1 (CDC34)	28
VHL	20.3
N8-E2 (Ube2M)	20
EloB	13.1
RBX1	12
EloC	10.8
Me-Ub	8.7
N8	8.6
Ub	8.6

The bands that can be labeled with certainty, as the reviewer pointed out, are UBE1 (118 kDa) and UBE2R1 (28 kDa), which we updated in **Supplementary Fig. 1 k, m**. The ubiquitinated IRE1 species are in the MW range that overlap with CUL2 and UBA3, therefore we only point out that the monoubiquitinated species are in the range of 50 – 100 kDa.

15) Supplementary Figure 1i: Could the authors also highlight any lysines which did not show evidence of being modified by UBE2R?

Please see highlighted lysines on IRE1.KR in our reply to major comment 5) above and in **Supplementary Fig. 7 a,b**.

16) Page 8: Could the authors mention in the main text that the cryo-EM complex was cross-linked with BS3, and why this was necessary? This is nicely justified in the

materials and methods section (page 20 line 620), but this could be brought to the main text. Was the sample used for crosslinking and cryo-EM grid preparation the fully-formed SEC product?

Yes, we collected fractions corresponding to the IRE1:G6374:VCB ternary complex from SEC for crosslinking and cryoEM grid preparation – we have edited the **Materials and Methods** section to make this clear to readers. We have also included the rationale for using BS3 in the main text.

17) Out of curiosity, why is it unexpected that VHL engages the B2B dimer (page 8 line 228)?

Based on the gel filtration results from the current **Fig. 2d**, the stoichiometry at which VCB was engaging IRE1 was unclear to us. We had assumed that a monomeric IRE1 was arranged in such a manner on VHL that it would allow for targeting both K704/717. In hindsight, it should have been obvious to us. We have updated the text accordingly.

18) According to Supplementary Figure 3e the cryo-EM volume appears to have a good range of distributions. Would it be possible to include the directional FSCs and the sphericity values from the 3D FSC server? <https://3dfsc.salk.edu> Many 2D classes are shown, would it be possible to show fewer but to enlarge these so they can be better appreciate? For additional cryo-EM data validation, the authors could consider including the EMRinger score.

Thank you for suggesting map evaluation using the 3DFSC server. We have compared the directional FSC curve and the sphericity value from the server (left) to the conical FSC (cFSC) and conical FSC Area Ratio (cFAR) from CryoSPARC (right), which agree with each other. We added the cFSC and cFAR from CryoSPARC to **Supplementary Fig. 3** for consistency. We added the EMringer score to the updated **Supplementary Table 2**.

The 2D classes in **Supplementary Fig. 3a** are the ones we selected for subsequent data processing. We enlarged some classes from the 2D class examples for readers to see more clearly.

19) Rather than removing EloB altogether, residues 68-73 are well-resolved with density for side-chains and could be included in the deposited atomic model, to help show unambiguously that EloB is present in the sample. However, certain parts of VHL and EloC are not visible in the supplied sharpened map and therefore could be truncated from the atomic model.

We edited the model as requested: residues 68-73 are now added for EloB. We also stubbed side chains where density is poor for VHL and EloC. The statistics in **Supplementary Table 2** are updated as well. We have updated our model in PDB under the same accession code, 9N88, which will be released to public upon manuscript acceptance.

20) In Figure 4b-c-e-f, the figure legend states that the NanoBRET data for the wild-type IRE1 is the same as is shown in Fig. 2. Could the authors please clarify whether the experiment with the wild-type IRE1 was performed at the same time/in parallel to the mutants, to ensure that the lack of signal from the IRE1 mutants is due to the mutations themselves rather than any experimental issue, and to ensure a fair comparison between the wild-type and mutants?

No, transfection of WT was not performed at the same time as the mutants. To verify the consistency of the NanoBRET assay, we performed 3 replicates for each IRE1 construct/mutant, which were transiently transfected on different days, and showed that the results are internally consistent.

21) Page 9 line 280. In order to make strong claims such as ‘dramatically reduced’, it would be good to include whether the difference between the DC50 and Dmax values are statistically significant? If there does exist a statistically significant difference, then a more scholarly term such as ‘significantly reduced’ could be included.

We have performed N=3 biological replicates and calculated the p-values for **Fig. 4h** (p=0.0271), which shows a statistically significant difference between WT and H579A.

22) Page 10. The authors may wish to acknowledge that the E2 in the model presented in Figure 5 is different to the one used in the *in vitro* ubiquitination assays (UBE2R1/Cdc34 versus UBE2R2/Cdc34b). A structure of VHL-EloB-EloC-(NEDD8)-CUL2-RBX1-UBE2R1-Ub has also been published (PDB: 8RX0).

Thank you for bringing this to our attention. We have replaced the modeling with 8RX0, which has the same E2 (UBE2R1) as in the *in vitro* ubiquitination assay. The overall model and conclusion are not affected.

23) Several typos/inconsistencies in the methods which should be corrected prior to publication.

Thank you for pointing this out. We have proof-read and corrected the methods section as needed.

24) As the authors will be aware, another IRE1 degrader was recently published. This should be appropriately referenced in the manuscript: <https://pubs.rsc.org/en/content/articlelanding/2025/md/d5md00028a>.

Thank you for referring to this paper. We also came across this paper, but didn't have a chance to add it to our reference list since it came out right at the time we submitted our manuscript. This publication discussed the development of CPD-2828 and some further SAR. The optimization based on CPD-2828 either made IRE1 degradation less efficient, or led to no difference. We have now cited this paper as well.

25) As the authors will be aware, cryo-EM structures of whole CRL2-VHL complexes with E2-Ub and neo-substrates have been recently disclosed and their relevant to ubiquitination specificity have been investigated by us (<https://doi.org/10.1126/sciadv.ado6492>) and the Schulman/Kleiger Labs

(<https://doi.org/10.1016/j.molcel.2024.01.022>), which should be duly discussed and cited.

We have incorporated the recommended reference in the **Discussion** section.

We agree to waive our anonymity in a spirit to enhance the transparency of the peer review process, Charlotte Crowe & Alessio Ciulli

Thank you for your transparency, it is very much appreciated!

REVIEWERS' COMMENTS

Reviewer #1 (Remarks to the Author)

The authors have satisfactorily addressed all the concerns raised in the previous reviews and have conducted the additional experiments as requested by the reviewers, significantly strengthening the manuscript and increasing the robustness of the results and the conclusions.

In the opinion of this Reviewer, the manuscript is ready for publication in Nature Communications after two minor additions.

We would like to thank the reviewer once more for the constructive suggestions.

Which high-throughput degradation assay was used to identify G6374 as the most potent PROTAC? The results (data) from this experiment should be included in the Supporting Information, at least in the form of a graphic showing the degradation level achieved by each degraded synthesised in the library. Moreover, the methodological aspects of this experiment should be included in the Materials and Methods Section.

We have added the requested methodological information to the Methods section.

The focus on the present manuscript is on the mechanistic and structural aspects of IRE1 degradation by the VHL-based PROTAC, G6374. A more detailed description of the identification and optimization of IRE1 degraders is beyond the scope of this present manuscript, and will be the subject of a dedicated future publication.

In parallel, the authors used in this revision a Fluorescence Polarization experiment to show cooperativity. It would be useful for the readers to mention in the manuscript if this biophysical approach has been used for this purpose, and add, if possible, references.

Cooperativity measurement of PROTAC-induced ternary complexes by fluorescence polarization displacement assays has been shown by Popow *et al* in the KRAS PROTAC paper. We have now added this reference to our manuscript:

“For ternary complex cooperativity measurement, we considered that IRE1 can form dimers^{4,5,7,48}, which may bias immobilization-based SPR analysis. We therefore turned to a solution-based FP approach, which has previously been used to characterize cooperativity for a KRAS PROTAC⁴⁹. To this end, we measured G6374’s ability to displace a VHL-binding HIF-1 α probe in the absence or presence of IRE1.”

49. Popow, J. et al. Targeting cancer with small-molecule pan-KRAS degraders. *Science* 385, 1338–1347 (2024).

Reviewer #2 (Remarks to the Author)

Thank you both for the constructive suggestions that made our manuscript stronger!

Reviewer #3 (Remarks to the Author)

The authors have rigorously addressed all my concerns and I am now happy to recommend publication. Congratulations on the nice body of work!

We greatly appreciate your positive assessment of our work!

Reviewer #4 (Remarks to the Author)

The authors have taken on board all comments, performed all of the suggested additional experiments and analyses, and have rigorously addressed all point made in their response and also in the main manuscript. The manuscript and the claims made therein have been significantly strengthened.

Notably, the degradation data has been strengthened by including timecourse treatments and biological replicates with statistical analyses. Further mode of action validation has been performed with the cis-VHL analogue PROTAC and siRNA of VHL, showing a VHL-dependent mechanism of degradation [this is in addition to the previous mode of action validation showing CRL and UPS-dependence]. A new proteomics experiment shows kinase selectivity. Another excellent addition is the biophysical characterisation (by SPR and FP) of the binary and ternary complexes, which shows that the PROTAC ternary complex is indeed cooperative.

The cryo-EM structure is of excellent resolution and coupled with the ubiquitination assays provides deep insight into the mechanism of IRE1 ubiquitination. The sample preparation for the cryo-EM has been clarified and brought up-front, which should help and guide the TPD community with solving more degrader ternary complexes. Additional studies with UBE2D1 and UBE2D2 have been performed and the results have been compared and contrasted with UBE2R1. Site-directed mutagenesis of the ubiquitinated lysines has also been performed, and discussion on this topic has been provided. These additional data, in conjunction with the IRE1 monomer and B2B dimer modelling with the full CRL2 scaffold fully rationalise why certain residues on IRE1 are targeted for ubiquitination.

Overall, the paper is significant and reports an exciting new VHL-recruiting IRE1 degrader molecule. Furthermore, it provides structural and mechanistic insights into the ubiquitination of IRE1 by CRL2VHL, which should be of great interest to the TPD field and contributes to advancing our knowledge about ubiquitination in TPD. We recommend that this manuscript should be published.

In a spirit to enhance transparency in the peer review process, we agree to waive our anonymity. Charlotte Crowe & Alessio Ciulli

Thank you both for the constructive suggestions that made our manuscript stronger!